# The UFM1 E3 ligase recognizes and releases 60S ribosomes from ER translocons

Linda Makhlouf[1,4], Joshua J. Peter[2,4], Helge M. Magnussen[2,4], Rohan Thakur[2], David Millrine[2,3], Thomas C. Minshull[1], Grace Harrison[2], Joby Varghese[2], Frederic Lamoliatte[2], Martina Foglizzo[1], Thomas Macartney[2], Antonio N. Calabrese[1], Elton Zeqiraj[1✉] & Yogesh Kulathu[2✉]

Stalled ribosomes at the endoplasmic reticulum (ER) are covalently modified with the ubiquitin-like protein UFM1 on the 60S ribosomal subunit protein RPL26 (also known as uL24)[1,2]. This modification, which is known as UFMylation, is orchestrated by the UFM1 ribosome E3 ligase (UREL) complex, comprising UFL1, UFBP1 and CDK5RAP3 (ref. 3). However, the catalytic mechanism of UREL and the functional consequences of UFMylation are unclear. Here we present cryo-electron microscopy structures of UREL bound to 60S ribosomes, revealing the basis of its substrate specificity. UREL wraps around the 60S subunit to form a C-shaped clamp architecture that blocks the tRNA-binding sites at one end, and the peptide exit tunnel at the other. A UFL1 loop inserts into and remodels the peptidyl transferase centre. These features of UREL suggest a crucial function for UFMylation in the release and recycling of stalled or terminated ribosomes from the ER membrane. In the absence of functional UREL, 60S–SEC61 translocon complexes accumulate at the ER membrane, demonstrating that UFMylation is necessary for releasing SEC61 from 60S subunits. Notably, this release is facilitated by a functional switch of UREL from a 'writer' to a 'reader' module that recognizes its product—UFMylated 60S ribosomes. Collectively, we identify a fundamental role for UREL in dissociating 60S subunits from the SEC61 translocon and the basis for UFMylation in regulating protein homeostasis at the ER.

UFM1 is a small ubiquitin-like modifier (UBL) that is ubiquitously expressed in most eukaryotes. In a pathway analogous to ubiquitin and other UBLs, post-translational attachment of UFM1 to the lysine residues of target proteins, UFMylation, is catalysed by an enzymatic cascade of E1 (UBA5), E2 (UFC1) and E3 (UFL1) enzymes[4–7]. The importance of UFMylation to human health is underscored by hypomorphic loss-of-function mutations in the UFMylation machinery that result in various pathologies, including cerebellar ataxia, neurodevelopmental disorders and skeletal abnormalities[8]. The main target of UFMylation is the 60S ribosomal-subunit protein RPL26[1] on ER-bound ribosomes. Given the proximity of RPL26 to the SEC61 translocon, RPL26 UFMylation has been suggested to regulate the biogenesis of secretory or membrane proteins at the ER membrane[1,2,9]. Other studies revealed that RPL26 UFMylation is stimulated by ribosome stalling and point to a role for UFMylation in the elimination of translocon-stalled ER nascent chains[2]. Although the precise function of RPL26 UFMylation is unclear, recent research demonstrated that it is catalysed by an unusual E3 ligase complex, which requires in addition to UFL1, the subunits UFBP1 (also known as DDRGK1 or c20orf116) and CDK5RAP3 (ref. 3). This heterotrimeric E3 ligase complex, hereafter UREL, is anchored to the ER membrane through the N-terminal transmembrane region of UFBP1. UREL lacks conserved catalytic features or domains that are commonly found in ubiquitin and UBL E3 ligases and uses a scaffold-type mechanism to bind to charged UFC1-UFM1 before transferring UFM1 onto a substrate lysine[3]. CDK5RAP3 is believed to be the substrate adaptor of UREL that inhibits in vitro UFMylation in the absence of ribosomes but contributes to selective mono-UFMylation of RPL26. How this unusual E3 ligase complex recognizes and specifically modifies the ribosome is unclear. Moreover, the functional implications of ribosome UFMylation remain enigmatic. To address these questions, we trapped the E3 ligase complex in a state poised to transfer UFM1 onto 60S ribosomes and describe the cryo-electron microscopy (cryo-EM) structure of this E3 ligase–ribosome complex. We also determined crystal structures of the UFC1–UFM1 conjugate, which mimics the charged E2 intermediate and the minimal catalytic module of UFL1–UFBP1 in a complex with UFC1. Our structure and function analyses reveal the mechanism of this unusual E3 ligase and point to a function of the ligase in the release of stalled and terminated ribosomes from the SEC61 translocon.

## UREL UFMylates 60S ribosomes

RPL26—a component of the 60S ribosomal subunit—is the main substrate of UREL and is modified on two adjacent lysine residues, Lys132 and Lys134 (refs. 1,2). However, it is unclear whether UREL recognizes ribosomes in the context of 60S, 80S or polysomes to UFMylate RPL26.

[1]Astbury Centre for Structural Molecular Biology, School of Molecular and Cellular Biology, Faculty of Biological Sciences, University of Leeds, Leeds, UK. [2]MRC Protein Phosphorylation and Ubiquitylation Unit, University of Dundee, Dundee, UK. [3]Present address: Translational Immunology, Cancer Biomarker Centre, Manchester CRUK Institute, Manchester, UK. [4]These authors contributed equally: Linda Makhlouf, Joshua J. Peter, Helge M. Magnussen. ✉e-mail: e.zeqiraj@leeds.ac.uk; ykulathu@dundee.ac.uk

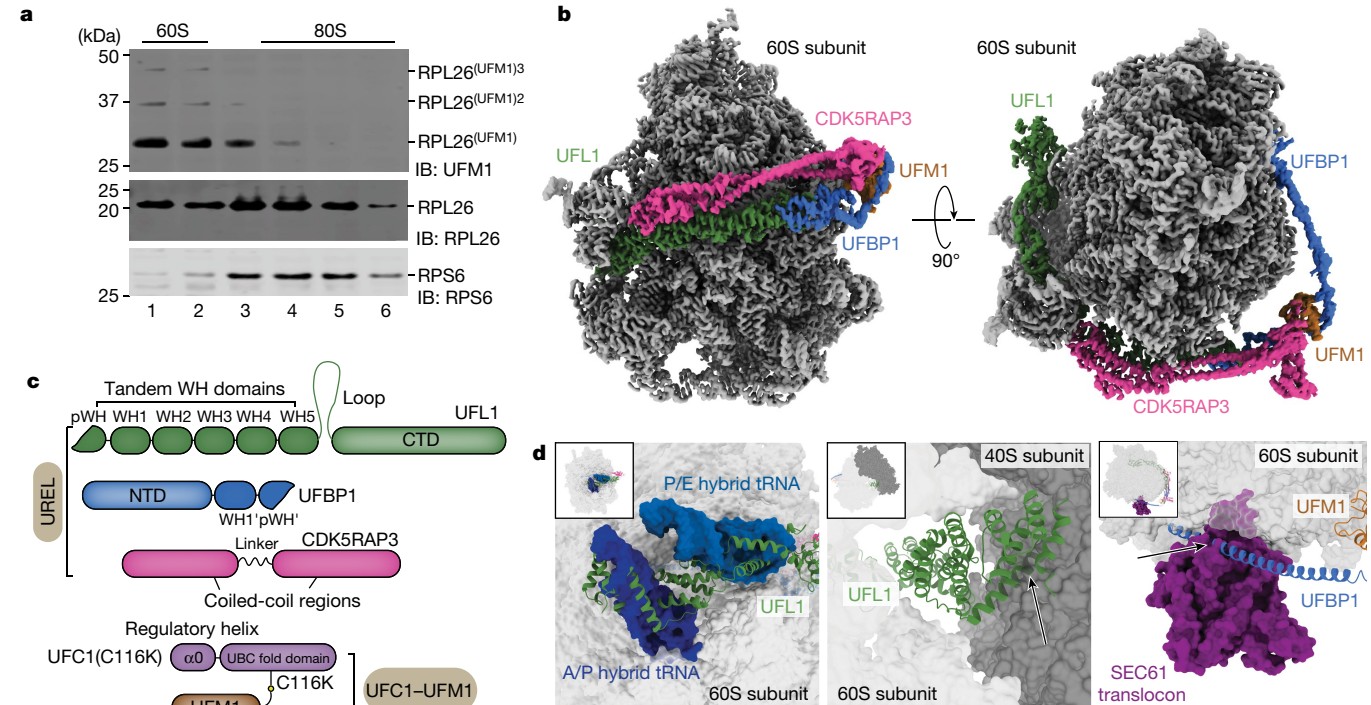

**Fig. 1 | Cryo-EM structure of the 60S–UREL–UFC1–UFM1 complex.**
**a**, ER-associated 60S ribosomal subunits are preferentially UFMylated in cells. The membrane fractions from WT HEK293 cells were solubilized and layered over a 10–30% sucrose gradient to separate individual ribosomal subunits. The 60S and 80S fractions were analysed for RPL26 UFMylation by immunoblotting (IB) using the indicated antibodies. The blot is representative of $n = 3$ independent experiments. Superscript values [2] and [3] refer to di- and tri-UFM1 RPL26 modifications, respectively. **b**, Composite cryo-EM density of the UREL ligase complex and UFM1 bound to the 60S ribosome. The 60S ribosome map is coloured in grey, and the UREL and UFM1 density map is coloured by protein as shown in **c**. **c**, Schematic of the domain architecture of UREL components (UFL1, UFBP1, CDK5RAP3) and the UFC1–UFM1 mimic. WH, winged-helix domain. NTD, N-terminal domain. **d**, Superpositions of UREL with published ribosome structures highlighting clashes with 60S ribosome-binding components. Left, the UFL1 C-terminal domain (CTD) occupies A-, P- and E-tRNA-binding sites and clashes with superimposed A/P and P/E tRNA (Protein Data Bank (PDB): 6W6L). Middle, the UFL1 CTD clashes with the 40S subunit (PDB: 6IP8). Right, the UFBP1 helix clashes with the SEC61 translocon (PDB: 6R7Q). The arrows indicate clashes.

We therefore analysed membrane fractions of HEK293 cells on sucrose density gradients and found that most of the RPL26 UFMylation was detected in the 60S fractions, with minor UFMylation also detected in the 80S fractions (Fig. 1a). Although this suggests that free 60S subunits may be the preferred substrate, we cannot rule out that 80S subunits undergo UFMylation before being split into 60S and 40S subunits. Thus, we reconstituted ribosome UFMylation in vitro using purified E1, E2 and E3 components to which purified 60S or 80S was added. UREL UFMylated 60S subunits more efficiently compared with 80S (Extended Data Fig. 1a). Furthermore, when UREL was incubated with a mixture of 60S and excess 80S ribosomes and analysed by sucrose density gradients, we found that UREL associates preferentially with 60S subunits (Extended Data Fig. 1b). Moreover, when ribosome stalling was induced in cells treated with anisomycin, we found an increase in RPL26 UFMylation and the association of UFL1, UFBP1 and CDK5RAP3 with UFMylated 60S ribosomes (Extended Data Fig. 1c,d). Collectively, these results suggest that UREL recognizes and UFMylates free 60S subunits. Furthermore, it also suggests that UREL remains associated with UFMylated 60S ribosomes. Mass spectrometry (MS) analyses of 60S ribosomes UFMylated with UREL in vitro revealed that Lys134 of RPL26 is the major site of modification (Extended Data Fig. 1e–h), highlighting the considerable substrate specificity of UREL.

## Overall structure of the 60S–UREL–UFC1–UFM1 complex

To understand how UREL recognizes 60S ribosomes and specifically modifies a single lysine residue on it, we aimed to trap UREL in a state in which it is positioned to transfer UFM1 from the E2 (UFC1) onto RPL26.

We engineered a stable mimic of the UFC1–UFM1 thioester bond by replacing the active-site cysteine residue of UFC1 with a lysine to generate an isopeptide bond between the C terminus of mature UFM1 (Gly83) and the ε-amino group of the introduced lysine (UFC1(C116K))[10] (Extended Data Fig. 1i,j). Isopeptide-linked UFC1–UFM1 binds to and crucially is stable in the presence of UREL (Extended Data Fig. 1k).

We imaged the stabilized UREL–UFC1–UFM1–60S ribosome complexes (Extended Data Fig. 1l,m) using cryo-EM (Extended Data Fig. 2 and Extended Data Table 1). UREL forms a clamp-like C-shaped architecture that wraps around the 60S subunit contacting the 'A site' at one end and a site close to the peptide exit channel at the other end (Fig. 1b,c). We did not observe any evidence of UREL bound to the 80S ribosome in our samples. Superposition of the 60S–UREL complex with 80S shows the C terminus of UFL1 positioned at the interface of the 60S and 40S subunits. Clashes with UFL1 would probably obstruct association of the 40S subunit with UREL-bound 60S (Fig. 1d), which is consistent with our biochemical data showing that UREL preferentially binds to 60S ribosomes over 80S. Notably, the C-terminal helical bundle of UFL1 occludes the aminoacyl (A), peptidyl (P) and exit (E) tRNA-binding sites (Fig. 1d). These observations suggest that UREL preferentially recognizes 60S ribosomes with vacant tRNA-binding sites or, alternatively, can disrupt tRNA binding to ribosomes.

## The architecture of UREL

We divide the clamp-like architecture of UREL into four major modules: (i) the peptidyl transferase centre (PTC) module made up of the C-terminal regions of UFL1 that interact with binding sites for A, P and

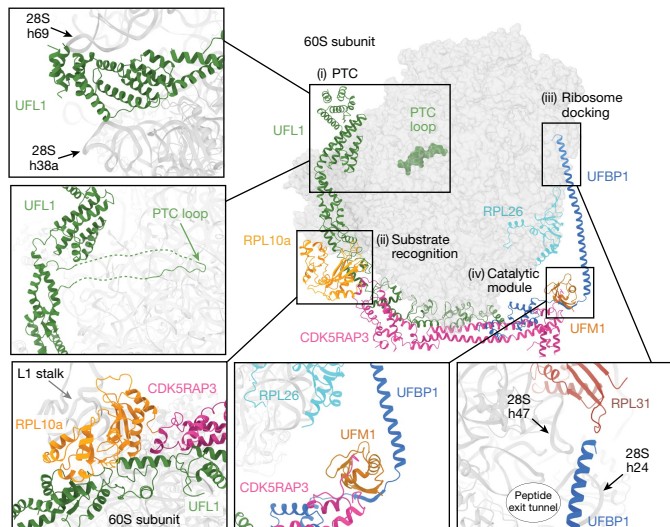

**Fig. 2 | Summary of key interacting modules of the 60S–UREL–UFC1–UFM1 complex.** UFL1 interacts with the PTC (i). Top, the UFL1 C-terminal domain sits within the tRNA-binding groove and is clamped between 28S rRNA helices 69 (h96) and 38a (h38a). Bottom, the UFL1 loop (residues 436–448; solid line) reaches into the PTC and binds proximal to the P-site. Disordered connecting residues are shown as dashed lines. The substrate-recognition module (ii). UFL1 and CDK5RAP3 bind to RPL10a from the L1 stalk. The ribosome docking module (iii). The UFBP1 helix docks onto the ribosome adjacent to the peptide exit tunnel, near 28S rRNA helices 47 and 24, as well as RPL31. The catalytic module of the UREL complex (iv). UFM1 is bound by CDK5RAP3 and UFBP1, with UFM1 positioned directly above the C terminus of RPL26 near the site of UFMylation.

E tRNAs and the PTC; (ii) the substrate-recognition module made up of the central portion of UREL that mediates contacts with ribosomal proteins; (iii) the ribosome-docking sequence (RDS) sitting at the other end of the clamp that is made up of the N-terminal helical stretch of UFBP1; and (iv) the catalytic module, consisting of the regions of UREL that are responsible for UFC1–UFM1 binding and transfer of UFM1 onto RPL26 (Fig. 2).

The cryo-EM structure recapitulates our recent AlphaFold prediction of the UFL1–UFBP1 complex[3,11], confirming the presence of seven winged-helix (WH) domains, five from UFL1 (WH1–5), one from UFBP1 (WH1′) and a composite WH domain (pWH–pWH′). The partial WH domains from the N terminus of UFL1 (pWH) and the C terminus of UFBP1 (pWH′) come together to form this composite WH domain, thereby bridging the two proteins together[3,11]. This pWH–pWH′ association is mainly mediated by hydrophobic residues that glue the ligase complex together (Extended Data Fig. 3a).

## Mode of ribosome recognition by UREL

All WH domains except for UFBP1 WH1′ contact the ribosome (Extended Data Fig. 3b). UFL1 WH1 and WH2 contact RPL13 (eL13) where Asn78, Gln83 (WH1) and Gly150 (WH2) are in hydrogen-bond distance to RPL13 Gln115, Ser122 and Glu108, respectively (Extended Data Fig. 3c). The ligase forms an extensive interaction network with RPL10a (also known as uL1) of the ribosomal L1 stalk, a dynamic region that is involved in E-tRNA removal and small-subunit rotation during translation. UREL binding to RPL10a buries a surface area of around 1,075 Å[2], with binding predominantly mediated by UFL1 WH4 and WH5, as well as CDK5RAP3 (Extended Data Fig. 3d–f). Hydrophobic contacts between UFL1 WH4 residues Trp256, Phe260, Tyr266 and Leu275, CDK5RAP3 residues Val377, Leu378 and Val380, and RPL10a Phe189 are the main interactions at this interface (Extended Data Fig. 3e). We also observed hydrogen bonding between UFL1 Asn264, Glu268 and Glu270, RPL10a Thr52,

His184 and Lys47, and CDK5RAP3 Gln384 (Extended Data Fig. 3f). Moreover, UFL1 WH5 forms many charged interactions with RPL10a. Comparing the unbound and ligase-bound 60S density maps within the same sample reveals that RPL10a undergoes a substantial positional shift after binding of UFL1 and CDK5RAP3, stabilizing the L1 stalk into a more closed conformation (Supplementary Video 1).

UFL1 makes additional contacts with the ribosome, burying a total surface area of around 2,356 Å[2]. The UFL1 WHDs almost exclusively interact with ribosomal proteins rather than rRNA, contacting RPL36A, RPL36 and RPL13 (Extended Data Fig. 3b). Furthermore, the UFL1 C-terminal helical bundle contacts RPL11 and we observed 28S rRNA helices 69 and 38a engaging the C-terminal region of UFL1 (Fig. 2 and Extended Data Fig. 3b).

CDK5RAP3 forms the central part of the C-clamp, acting as a bridge that contacts both UFL1 and UFBP1 (Extended Data Fig. 3b,g–i). CDK5RAP3 anchors UREL onto the ribosome through a module that interacts with RPL10a, which we have termed the RPL10a-binding domain (RBD) (Extended Data Fig. 3g). In addition to the RBD, CDK5RAP3 contains an extended coiled-coil domain (CCD) that spans the WH repeats, and a UFM1- and UFBP1-binding domain (UUBD) (Extended Data Fig. 3g). Besides contacting the ribosome, all three domains of CDK5RAP3 make continuous contacts with one or more ligase subunits and UFM1 (Extended Data Fig. 3g), which aligns with the idea that CDK5RAP3 is an integral component of the E3 ligase complex. In addition to interactions with the ribosome, the RBD of CDK5RAP3 makes contacts with UFL1 WH2, WH3 and WH4, primarily through electrostatic interactions (Extended Data Fig. 3h). The CCD contacts UFL1 WH2 and a loop in UFBP1 WH1′ that contains the conserved 'DDRGK' motif (Extended Data Fig. 3i). Furthermore, UUBD of CDK5RAP3 interacts with UFBP1 WH1′ (Extended Data Fig. 3g). Moreover, CDK5RAP3 CCD contacts rRNA helix 16 and RPL13 (eL13). These extensive interactions of CDK5RAP3 with the ribosome and other ligase components suggest a functionally critical role for CDK5RAP3 in UFMylation. Consistent with this idea, deleting *CDK5RAP3* in HEK293 cells almost completely abrogated RPL26 UFMylation (Extended Data Fig. 3j).

The N-terminal region of UFBP1 forms a helical arm that contacts 28S rRNA helices 24 and 47 proximal to RPL31 (also known as eL31) and we name this region the UFBP1 RDS (Fig. 2 and Extended Data Fig. 3k). Notably, SEC61, the signal-recognition particle (SRP) and the SRP receptor also bind to this region of the ribosome[12,13].

## PTC remodelling by UFL1

We noticed additional density in the P-site of the PTC that could not be attributed to the ribosome. Importantly, this additional density is not present in our unbound 60S map (Fig. 2 and Extended Data Fig. 4a). We also do not see density for nascent peptide within the peptide exit tunnel or any major conformational changes to the peptide exit tunnel, including tunnel proteins RPL4 (also known as uL4), RPL17 (also known as uL22) and RPL39 (also known as eL39). Close inspection of the cryo-EM density enabled us to model a 13-residue segment (UFL1(436–448)) that is part of an 88-residue UFL1 protrusion (UFL1(388–476)) from the C-terminal helices, which we call the UFL1 PTC loop (Extended Data Fig. 4b).

Notably, the PTC loop appears to remodel key translation elongation and termination bases involved in PTC function[14]. 28S rRNA A4548 rotates towards the P-site to stack with the aromatic ring of UFL1 Tyr443 (Extended Data Fig. 4c). Moreover, U4452 flips around 90° to be in proximity to UFL1 Gly437, which partially occludes the A-site (Extended Data Fig. 4d). Importantly, A4548 is a highly mobile base within the PTC that aids in tRNA positioning for peptide-bond formation and peptide release, whereas U4452 is involved with A-tRNA binding[14–17]. We also observed A3908 and A4385 stacking with UFL1 Asn439 and Arg441, respectively[18] (Extended Data Fig. 4e,f). As these bases are critical for translational elongation and termination, the intricate interactions

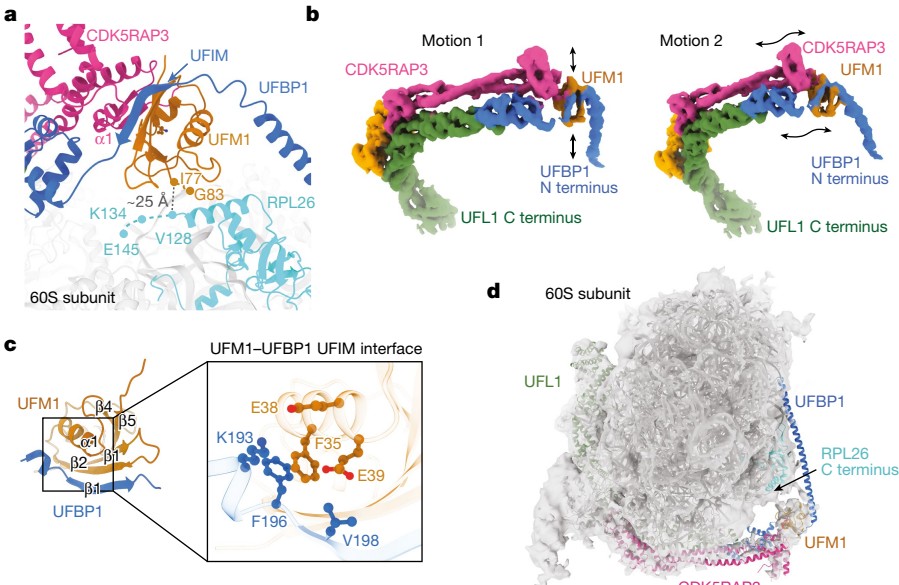

**Fig. 3 | The UFIM motif of UFBP1 binds to UFM1. a**, View of the catalytic region of the UREL complex bound to the 60S ribosome showing the UFBP1 UFIM and CDK5RAP3 N terminus bound to UFM1. The distance between the C termini of UFM1 and RPL26 in our cryo-EM model is shown as a black dashed line. Missing residues of UFM1 and RPL26 are shown as dashed lines in their respective colours. **b**, 3DFlex was used to generate motion models of the UREL ligase complex. The black arrows indicate the direction of motion observed. **c**, AlphaFold prediction of the UFBP1 UFIM–UFM1 complex. Inset: an enlarged view of the UFBP1 β1 and UFM1 β2 interface; key interactions are highlighted. **d**, Cryo-EM density map of UFMylated 60S ribosomes shown in transparent grey with a model of UREL ligase bound to 60S rigid-body fitted into the density.

and remodelling of the PTC suggest that interactions of the UFL1 PTC loop with the PTC may be a detection mechanism for translationally terminated ribosomes or may even suggest UFMylation-independent functions for UFL1 (ref. 18).

## UREL binds to UFMylated 60S through the UFIM motif

UREL wraps around the ribosome in such a way that the catalytic module, located at the intersection of UFL1, UFBP1 and CDK5RAP3, orients the C-terminal Gly83 of UFM1 towards the stretch of RPL26 that contains Lys134, the major site of UFMylation (Fig. 3a). However, density for the C termini of RPL26 and UFM1 are not visible in our cryo-EM maps and there is a distance of approximately 25 Å between RPL26 and the ligase-bound UFM1. To test whether there is flexibility in UREL, we performed 3DFlex analysis[19]. Indeed, while most of UREL remains stably bound, significant movements were observed around the UREL catalytic module, particularly at the juncture of UFL1, UFBP1 and CDK5RAP3 where UFM1 density is present (Fig. 3b and Supplementary Video 2). This plasticity suggests that the catalytic module can adopt a range of conformations that may be necessary for efficient UFMylation.

We find UFM1 to be 'sandwiched' between the WH1' domain and the long N-terminal helix of UFBP1. A ten-residue segment (UFBP1(196–205)) connecting the WH1' and N-terminal domains of UFBP1 binds to UFM1. A AlphaFold Multimer[20] prediction revealed that this linker forms a β-strand when bound to β2 of UFM1 (Fig. 3c). This motif and mode of UFM1 binding resembles the LC3-interacting region (LIR) motif of UBA5 binding to UFM1 that has been termed the UFM1-interacting motif (UFIM)[21]. Additional interactions with UFM1 are mediated by the UUBD of CDK5RAP3. However, the local resolution of this region was around 4–5 Å, which prevented us from seeing discernible side chain density to unambiguously locate interacting residues. To our surprise, despite having a stable UFC1–UFM1 mimic in our sample, we did not see clear density for UFC1. As UFM1 is positioned in such a way that its C terminus is pointing towards RPL26, one interpretation is that our structure represents a post-UFMylated state in which the catalytic module is bound to UFMylated ribosomes.

We therefore generated UFMylated 60S ribosomes after addition of E1, E2, UREL and ATP in vitro and imaged the complex using cryo-EM. Indeed, the density map revealed that UFM1 is positioned similarly, with UREL bound to UFM1 (Fig. 3d). These results strongly imply that the ligase complex can bind to UFMylated ribosomes, and our structures represent post-UFMylated states.

Isothermal titration calorimetry (ITC) measurements revealed that a UFBP1 peptide containing just the UFIM motif (residues 178–204) binds to UFM1 with 2.2 μM affinity (Extended Data Fig. 5a). This suggests that the UFIM competes for binding to UFM1 on UFC1–UFM1 and may therefore inhibit UFMylation. Consistent with this idea, individual mutations of Lys193, Phe196 and Tyr198 within the UFIM motif designed to disrupt UFIM–UFM1 binding resulted in enhanced RPL26 UFMylation (Extended Data Fig. 5b). In summary, the affinity of the UFIM for UFM1 probably explains why our cryo-EM structure represents a UFMylated ribosome that is stably bound by the ligase complex.

## Catalytic mechanism
### E3 ligase–UFC1 binding

To understand the catalytic mechanism of UREL, we first investigated how the ligase complex binds to UFC1. We determined the crystal structure of a minimal region of UFL1–UFBP1 that is stable and catalytically competent (E3$_{mUU}$(ΔUFIM)) in a complex with UFC1 (Fig. 4a and Extended Data Table 2). Comparison of the cryo-EM and crystal structure shows that UFL1 WH1 and WH2 superimpose well with a root mean squared deviation of 0.98 Å (Extended Data Fig. 5c). The composite pWH–pWH' domain and UFBP1 WH1' show almost identical folds in the two structures. However, in the cryo-EM model, these two domains are seemingly pulled towards the CDK5RAP3 CCD through the DDRGK loop in UFBP1 (Extended Data Fig. 5d). Moreover, the crystal structure reveals an N-terminal helix (α1) of UFL1, which is connected to pWH through a hinge. Importantly, this helix mediates multiple hydrophobic interactions with UFC1 α2 (Fig. 4b). Single point mutations in UFC1 (L32R, I40R) or UFL1 (I8R, F15R, Q19R) designed to disrupt this helical interface abolish complex formation between UFC1 and E3$_{mUU}$(ΔUFIM)

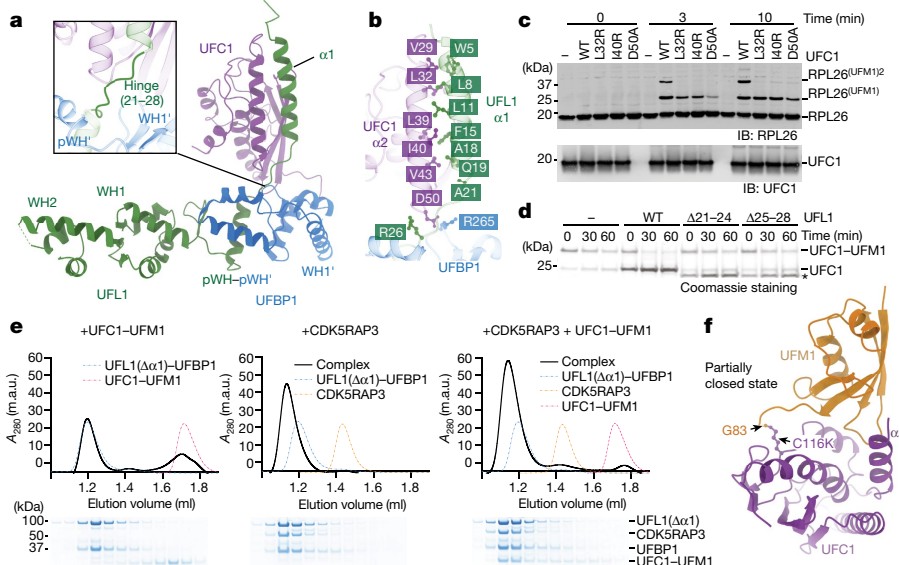

**Fig. 4 | The mechanism of E2 recognition and UFMylation. a**, The crystal structure of E3$_{mUU}$(ΔUFIM) bound to UFC1 shown as a cartoon representation. Inset: enlarged view of the hinge region that connects UFL1 WH1 to α1. **b**, Enlarged view of the interaction between UFC1 α2 and UFL1 α1. **c**, Immunoblot analysis of in vitro 60S ribosome UFMylation in the presence of the indicated E2 mutants that are defective in E3 binding. The blot is representative of $n = 3$ independent experiments. **d**, Lysine-discharge assays in the presence of E3$_{mUU}$ bearing truncations in the hinge region. The asterisk indicates UFL1(Δ21–24) and UFL1(Δ25–28). The Coomassie-stained SDS–PAGE gel is representative of $n = 3$ independent experiments. **e**, CDK5RAP3 and UFL1 form a composite

binding site for the UFC1–UFM1 conjugate. Top, SEC elution profiles of UFL1(Δα1)–UFBP1–UFC1–UFM1, UFL1(Δα1)–UFBP1–CDK5RAP3 and UFL1(Δα1)–UFBP1–CDK5RAP3–UFC1–UFM1 complexes. Bottom, the corresponding peak fractions were separated on a 4–12% SDS–PAGE gel under reducing conditions and Coomassie stained. $A_{280}$, absorbance at 280 nm. **f**, The crystal structure of the UFC1–UFM1 conjugate shown as a cartoon representation. UFC1 is coloured in purple, UFM1 is coloured in orange and the isopeptide bond formed between UFM1 Gly83 and UFC1(C116K) is highlighted and shown as a ball and stick representation.

and impact ribosome UFMylation without altering the association between UFL1 and UFBP1 in E3$_{mUU}$(ΔUFIM) (Fig 4c and Extended Data Fig. 5e–k). Importantly, these mutants are also defective at aminolysis, highlighting an important role for E2 interaction and activation through UFL1 α1 (Extended Data Fig. 5l).

While UFL1 α1 binds to UFC1 in the crystal structure, no clear density is visible for UFL1 α1 in the cryo-EM structure. As the composite pWH–pWH′ domain adopts a similar conformation in our structures, it suggests that the hinge region that connects UFL1 pWH and α1 is flexible and enables UFL1 α1 to adopt multiple conformations to optimally position UFC1–UFM1 for catalysis. Indeed, truncating the hinge by deleting four-amino-acid stretches abolishes the ability of E3$_{mUU}$ to activate UFC1–UFM1 for aminolysis (Fig. 4d). Furthermore, deletion of UFL1 α1 in E3$_{mUU}$(ΔUFIM) (E3$_{mUU}$(ΔUFIM-Δα1)) abrogates binding to both UFC1 and UFC1–UFM1 (Fig. 4e and Extended Data Fig. 6). However, in the presence of the UFIM motif, E3$_{mUU}$(Δα1) can bind to free UFM1 and, consequently, UFC1–UFM1 but not UFC1 on its own. These results further demonstrate the ability of UREL to bind to the UFC1–UFM1 conjugate solely through UFM1 as seen in our cryo-EM structure.

**Catalytic site of UREL**

In addition to the interactions with UFL1 α1, UFC1 contacts UFBP1 WH1′ and pWH′. In the E3$_{mUU}$(ΔUFIM)–UFC1 crystal structure, UFC1 Asp50 interacts with Arg265 of the DDRGK motif in UFBP1 WH1′. Although the UFC1(D50A) or UFBP1(R265A) mutations do not disrupt complex formation between UFC1 and E3$_{mUU}$(ΔUFIM), these mutants impair both aminolysis and ribosome UFMylation (Fig. 4c and Extended Data Fig. 5e,m,n). In the cryo-EM structure, UFBP1 Arg265 interacts with CDK5RAP3 CCD, making it unavailable for UFC1 binding. CDK5RAP3 may also produce steric clashes with UFC1 bound to UFL1 α1, therefore making the binding of UFC1 to this region unfavourable in the presence of CDK5RAP3. It is therefore possible that the purpose of

the UFC1-α2–UFL1-α1 interaction is to recruit UFC1–UFM1 into the UREL–ribosome complex before transfer to another region of the ligase where catalysis occurs.

To investigate this possibility, we performed cross-linking MS (XL-MS) to capture the interactions between UFC1–UFM1 and the UREL–60S ribosome complex1. The XL-MS data recapitulate many of the interactions that we observed in the cryo-EM structure. In addition, we observed two distinct binding sites for UFC1. We observed UFC1 to form cross-links with RPL26 and the helical arm of UFBP1. On the basis of these XL-MS data, we can position UFC1 at this site such that it faces RPL26 (Extended Data Fig. 7a–d), further strengthening the notion that UFC1–UFM1 is positioned at this site before catalysis. Furthermore, a substantial number of cross-links are observed between UFC1 and a region of CDK5RAP3 that is close to UFL1 WH1, the UFL1–UFBP1 interface pWH–pWH′ and UFBP1 WH1′, suggesting the presence of a composite binding site for UFC1. To demonstrate that a composite UFC1–UFM1 binding site made up of CDK5RAP3 and UFL1–UFBP1 exists, we took advantage of the observation that the affinity of UFL1(Δα1) for UFC1–UFM1 (Extended Data Fig. 6k) is insufficient for co-elution on size-exclusion chromatography (SEC) (Extended Data Fig. 7e). We posited that, if a composite binding site existed, then UFC1–UFM1 would act as a glue to bridge UFL1(Δα1)–UFBP1 and CDK5RAP3. Indeed, in the presence of CDK5RAP3, UFL1(Δα1)–UFBP1 forms a stable complex with UFC1–UFM1 (Fig. 4e) and CDK5RAP3 binds to a preformed E3$_{mUU}$–UFC1–UFM1 complex with a $K_d$ of 500 nM (Extended Data Fig. 7f–h). Taken together, our results suggest that UFC1–UFM1 binds to UFL1 α1 as seen in the crystal structure and to a second site on UREL, which is probably the site at which the transfer of UFM1 to RPL26 occurs.

**Active UFC1–UFM1 conformation**

UFL1–UFBP1 uses a scaffold-type E3 mechanism[3], and such ubiquitin or UBL E3 ligases typically activate the E2-Ub/UBL thioester by binding

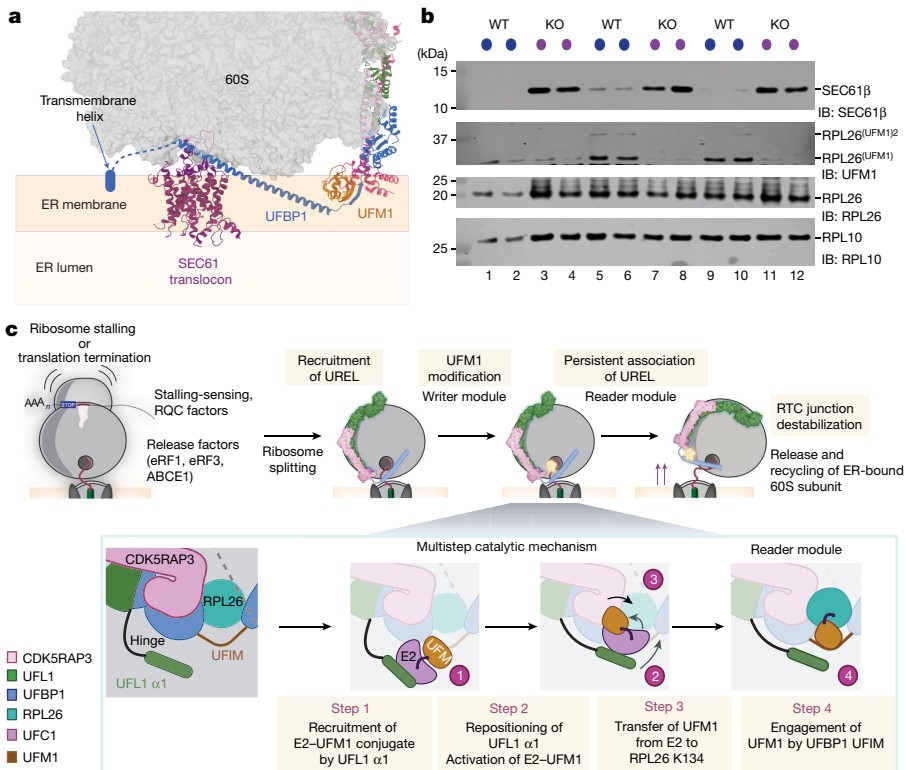

**Fig. 5 | UFMylation of RPL26 mediates dissociation of the 60S ribosomal subunit from the ER translocon. a**, UREL–UFM1 superimposed onto the structure of 80S–SEC61 (PDB: 6R7Q). Modelling UREL binding to SEC61 translocon-bound ribosomes suggests that the ligase complex will destabilize the 60S–SEC61 interaction. The approximate position of the ER membrane and ER lumen relative to the SEC61 translocon are shown. **b**, Loss of UFMylation leads to an accumulation of 60S ribosomes with the translocon. SEC61β association with 60S ribosomes was analysed in membrane fractions from parental WT and *CDK5RAP3*-KO cells. Lysates were normalized to uniform RNA concentration (using absorbance at 254 nm) and fractionated on 10–30% sucrose density gradients. The fractions corresponding to 60S from three independent replicates were analysed by immunoblotting using the indicated antibodies. **c**, Model for 60S–SEC61 translocon recognition by UREL, showing the multistep catalytic mechanism leading to RPL26 UFMylation and the subsequent conversion of UREL to a reader that splits the 60S ribosomal subunit from the translocon (see Discussion). The diagram in **c** was created using BioRender.

to and inducing a folded-back closed conformation that extends the C-terminal tail of Ub/UBL[10,22,23]. This closed conformation is often associated with the high-energy state of a loaded spring that would be released by cleavage of the thioester and formation of the isopeptide bond. To understand interactions between UFC1 and UFM1 independent of the UREL complex, we determined the crystal structure of the stable isopeptide-linked UFC1–UFM1 mimic (Fig. 4f). When compared to other E2–UBL structures in closed and open conformations[24], our structure reveals that UFM1 adopts an intermediate state (Extended Data Fig. 8a). This conformation is stabilized by multiple contacts between UFM1 and UFC1 α0, which is a helical extension beyond the UBC fold of UFC1 (Extended Data Fig. 8b). We recently showed that deletion of α0 on UFC1 increases its intrinsic reactivity and RPL26 UFMylation[3]. Taken together, these data suggest that UFC1 α0 may prevent UFM1 from readily adopting a closed conformation, therefore stabilizing an intermediate state. Indeed, in a closed UFC1–UFM1 conformation, modelled based on closed E2–Ub structures[10,23], UFC1 α0 clashes with UFM1. We therefore postulate that α0 would need to be remodelled during E3-mediated activation of UFC1. Superposition of the UFC1–UFM1 structure onto the cryo-EM structure suggests how UFC1 and UFM1 will have to be remodelled to transition into a closed conformation (Extended Data Fig. 8c–e).

## UFMylation dissociates SEC61 from 60S

The N-terminal helical stretch of UFBP1 is positioned on the 60S subunit close to the exit tunnel where it could sterically clash with SEC61 binding (Fig. 1d). Modelling the UREL complex on 80S–SEC61 structures suggests that UREL could reorient the SEC61 complex at the membrane bilayer, thereby weakening the interaction between 60S and the translocon (Fig. 5a). This raises the possibility that a function of UFMylation could be to dissociate 60S from the SEC61 translocon after either normal termination of translation or after ribosome stalling during co-translational translocation at the ER. To explore this hypothesis, we analysed the co-sedimentation of SEC61 with 60S ribosomes in sucrose gradients of membrane fractions of HEK293 WT and *CDK5RAP3*-knockout (KO) cells. In contrast to WT cells, we observed substantial co-sedimentation of SEC61 with 60S subunits in the *CDK5RAP3*-KO cells (Fig. 5b and Extended Data Fig. 8f). To unequivocally demonstrate that UFMylation by UREL dissociates SEC61 from 60S, we set up an in vitro reconstitution assay using 60S–SEC61 complexes isolated from membrane fractions of *CDK5RAP3*-KO cells. These 60S–SEC61 complexes were incubated with UREL along with E1, E2 and free UFM1. To half of the mixture, ATP was added to enable UFMylation. UFMylation of RPL26 was observed in the 60S fractions and, consistent with its 'reader' function, UREL was enriched in UFMylated 60S fractions (Extended Data Fig. 8g). Importantly, SEC61 is dissociated efficiently from 60S only in samples in which UFMylation occurred and required the UFIM motif of UFBP1 (Extended Data Fig. 8h). This suggests that the presence of UREL by itself does not support splitting but requires UFMylation, whereby UREL transitions from being the 'writer' of the UFM1 modification to the 'reader' of UFMylated RPL26. Together, these results strongly support a role for UFMylation in freeing 60S from SEC61 at the ER and a role for UFMylation in ribosome recycling.

## Discussion

Ribosome UFMylation has been the subject of intense investigation and recent research has highlighted its importance in biological processes[1–3,25–28]. However, mechanistic insights on this fundamental process are lacking. On the basis of our analyses, we propose that ribosome UFMylation follows a series of ordered events. First, a tripartite interaction of UFL1 with the ribosome tRNA-binding sites, L1 stalk and the PTC mediates binding of UREL to 60S ribosomes. Further bridging interactions by CDK5RAP3 stabilize UREL binding to the ribosome, positioning the catalytic centre near RPL26. This then initiates a catalytic cycle which, as we propose, occurs through four distinct steps (Fig. 5c). In step 1, UFL1 α1 binds to charged UFC1 (UFC1-UFM1) and recruits it to the ribosome. While both UFL1 and the E1 (UBA5) bind to the same interface on UFC1, the higher affinity of UFL1 for UFC1-UFM1 (ref. 28) results in the transfer of UFC1-UFM1 onto the ligase complex. In step 2, UFC1-UFM1 is repositioned on the UREL catalytic centre and this transition is aided by the flexible hinge region connecting UFL1 α1 with WH1. In this catalytic region, the UFC1-binding site comprises a composite patch formed by UFL1, UFBP1 and CDK5RAP3 (Fig. 4e and Extended Data Figs. 5 and 7).

Once repositioned at this site, in step 3, UREL induces the formation of an active UFC1-UFM1 conformation, which probably involves remodelling of UFC1 α0. Subsequently, UREL catalyses the transfer of UFM1 from UFC1-UFM1 onto RPL26 Lys134. During this process, UFBP1 UFIM competes for binding to UFM1, which may explain the increased UFMylation observed after mutations within the UFIM–UFM1 interface (Extended Data Fig. 5b). In step 4, UFBP1 UFIM and CDK5RAP3 bind to UFMylated RPL26, which presumably further stabilizes UREL binding to 60S ribosomes. Crucially, the flexibility observed in the catalytic module of UREL (Supplementary Video 2) lends support to this multistep cycle which probably involves several conformational changes. Collectively, these features distinguish UREL from other multisubunit E3 ligases, therefore providing a unique 'transfer and stabilize' mechanism instead of a canonical 'transfer and release' mechanism.

Despite having set out to trap UREL in a state in which it is poised to transfer UFM1 from UFC1 onto RPL26, the UFBP1–UFIM–UFM1 interaction suggests that the 60S–UREL–UFC1–UFM1 cryo-EM structure represents the post-UFMylation state in which UFMylated RPL26 is stably bound by UREL. This may also explain why we do not observe any density for UFC1, as our structure is indicative of a state in which UFC1 is no longer required. A major question in the field is to identify factors that 'decode' or 'read' the UFM1 modification on ribosomes. Our results suggest that UREL is not only the modifier but also functions as a reader module that strongly binds to UFMylated RPL26. As the reader module, UREL is stably bound to ribosomes such that the long helix at the N terminus of UFBP1 is perfectly positioned to impede 60S–SEC61 interactions and mediate downstream functions of RPL26 UFMylation.

Functionally, UFMylation of 60S subunits at the ER by the UREL complex could impact ribosome quality control through several mechanisms. First, the binding of the C-terminal region of UFL1 to the tRNA-binding sites and the UFL1 loop that remodels the PTC suggests a mechanism in which detecting stalled or terminated ribosomes is important. Although the exact function of the PTC loop is unknown, the loop is reminiscent of the eRF1 peptidyl-tRNA hydrolysis module, which binds within the A-site to hydrolyse the P-site tRNA–peptide bond after translation termination[29,30] (Extended Data Fig. 8i). Notably, despite eRF1 and PTC loops occupying different tRNA sites, the apex of both proteins have residues GGQ (eRF1 catalytic motif) or GGN (UFL1 PTC loop)[31,32] (Extended Data Fig. 8i). It is therefore tempting to speculate that the PTC loop may also perform a catalytic role. Our results also suggest that, due to steric hindrance, binding of UFL1 to the 60S ribosome would be incompatible with simultaneous binding of the 40S subunit. It is therefore probable that UFL1 preferentially recognizes free 60S ribosomes. Listerin (LTN1) is a ubiquitin E3 ligase and

a component of ribosome quality control that binds to 60S ribosomes and ubiquitylates stalled nascent chains[33]. The two ligases bind to 60S in diametrically opposite ways (Extended Data Fig. 8j), and our work suggests that UREL binds to vacant 60S subunits, highlighting distinct quality-control functions for the two ligases[34,35].

At the other end of UREL, the long N-terminal helix of UFBP1 that is positioned at the exit tunnel could destabilize the 60S–SEC61 translocon interaction, thereby releasing terminated 60S ribosomes. Alternatively, working with the ribosome quality-control machinery, UFMylation could occur on ribosomes that stall during co-translational translocation of secretory and membrane proteins at the ER membrane, serving as a signal to remove stalled nascent peptides that have been inserted into the translocon[36]. As such, an important function of the multiple interactions of the C terminus of UFL1 could be a failsafe mechanism to ensure that translation is not reinitiated at these ribosomes by clashing with the 40S, blocking the tRNA-binding sites and remodelling the PTC. Thus, a functional outcome of UFMylation that also requires the function of UREL as a reader may be to recycle terminated or stalled translocon-bound 60S subunits.

Several groups have identified ribosomes to be stably associated with the translocon after termination of protein synthesis and it is unclear how they are separated and recycled[37,38]. Our research answers this long-standing question. After dissociation from SEC61, stable binding of UREL to UFMylated 60S would also serve a secondary function to protect these 60S subunits from re-engaging in translation at the ER. In this scenario, de-UFMylation by UFSP2 would be a critical step to release UREL, therefore freeing 60S subunits from the ER. This fundamental role for UFMylation in the recycling of 60S subunits from the ER may explain why the UFMylation machinery is conserved in most eukaryotes and why the loss of UFMylation is detrimental to organism fitness and survival. Our study also raises an important question of how ribosome recycling is achieved in eukaryotes such as fungi, which lack UFMylation, hinting at the existence of an alternative pathway. Overall, we identify a unique E3 ligase mechanism that helps to understand how UFMylation regulates a fundamental step in protein synthesis at the ER membrane. It is remarkable that a completely dedicated post-translational modification with its own molecular machinery has evolved to mainly modify a single lysine residue in cells, the outcome of which is to free and recycle 60S ribosomes from ER translocons, therefore maintaining proteostasis at the ER.

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

# Methods

## Antibodies and recombinant proteins
Details of the antibodies and recombinant proteins used in this study are provided in Supplementary Table 1.

## Mammalian cell culture and cell line generation
Flp-In T-REx HEK293 cells (Invitrogen, R78007) were cultured in high-glucose DMEM supplemented with 10% (v/v) fetal bovine serum (FBS), 50 mg ml$^{-1}$ penicillin–streptomycin and 2 mM L-glutamine. Cells were maintained at 37 °C under 5% $CO_2$ in an incubator in a humidified environment and routinely checked for mycoplasma. *CDK5RAP3*-KO cells were generated using CRISPR–Cas9. CRISPR sense and anti-sense guides were cloned into pX335 (DU64982) and pBABED puro U6 (DU64977) plasmids, respectively. In a separate strategy, single guide RNAs were cloned into the px459 vector (Addgene, 48139). In brief, around 2 million cells were seeded into a 10 cm dish in antibiotic-free Dulbecco's modified Eagle medium (DMEM) and transfected with 1 μg plasmid DNA using Lipofectamine 2000 (Invitrogen, 1168019) according to the manufacturer's instructions. Then, 24 h after transfection, cells were selected in 2 μg ml$^{-1}$ puromycin for 24 h followed by a 24 h recovery period in preconditioned medium. Cells were submitted for single-cell sorting, expanded and knockouts were confirmed by sequencing and immunoblot analysis.

## Cytosolic and membrane fractionation
For chemical induction of ribosome stalling, cells were treated with 200 nM anisomycin for 4 h before collection. The parent cell line (Flp-In T-REx HEK293 cells) and KOs were washed once in ice-cold PBS, collected in ice-cold PBS and pelleted by centrifugation at 800$g$. Cell pellets (around $2 \times 10^6$ cells) were resuspended in 125 μl of 0.02% (w/v) digitonin, 50 mM HEPES pH 7.5, 150 mM NaCl, 2 mM $CaCl_2$, and 1× cOmplete protease inhibitor cocktail EDTA-free (Roche). Cells were incubated on ice for 10 min and centrifuged at 17,000$g$ for 10 min at 4 °C. The supernatant was transferred to a fresh tube (cytoplasmic extract). The remaining pellet was washed with 1× PBS and centrifuged at 7,000$g$ for 5 min at 4 °C. The pellet was resuspended in 125 μl 1% Triton X-100, 50 mM HEPES pH 7.5, 150 mM NaCl and 1× EDTA-free protease inhibitor cocktail tablet. This was further incubated on ice for 10 min and centrifuged at 17,000$g$ for 10 min at 4 °C. The supernatant was transferred to a new Eppendorf tube (membrane extract). Equal volumes of the cytosolic and membrane fractions were resolved by SDS–PAGE and analysed using immunoblotting.

## Expression and purification of recombinant proteins
UBA5, UFC1, UFM1, UFL1–UFBP1 and CDK5RAP3 were expressed and purified as described previously[3]. GST-3C-UFBP1(178–204) was applied onto Glutathione Sepharose 4B beads (Cytiva) followed by 3C-protease cleavage on the beads and purified on the HiLoad 26/600 Superdex 75 pg column, pre-equilibrated with 25 mM HEPES pH 7.5 and 200 mM NaCl. E3$_{mUU}$ constructs were applied onto HisTrap columns, pre-equilibrated with 50 mM Tris pH 7.5, 200 mM NaCl, 20 mM imidazole (pH 8.0) and eluted with the same buffer containing 300 mM imidazole. Constructs with a cleavable His-tag were incubated with 1:50 TEV protease and dialysed against 25 mM Tris pH 7.5, 200 mM NaCl at 4 °C overnight and further purified on the HiLoad 26/600 Superdex 75 pg column, pre-equilibrated with 25 mM HEPES pH 7.5, 200 mM NaCl and 1 mM DTT. For crystallization of the E3$_{mUU}$(ΔUFIM)–UFC1 complex, 6×His-TEV-UFL1(1–179) was co-expressed with a UFC1-UFBP1 204-C fusion construct. Here, the His-tag was not cleaved.

## Discharge assays
Single-turnover lysine discharge assays were performed to analyse the activity of UFC1 and UFL1–UFBP1 as described previously[3]. In brief, UFC1 was charged by incubating 0.5 μM UBA5, 10 μM UFC1 and 10 μM UFM1 in reaction buffer containing 50 mM HEPES pH 7.5, 50 mM NaCl, 0.5 mM DTT, 10 mM ATP and 10 mM $MgCl_2$ for 20 min. The reaction was quenched by addition of 50 mM EDTA (pH 8.0) to the reaction mix followed by incubation for 10 min at room temperature. Discharge was performed in the presence of 50 mM lysine (pH 8.0). The reaction was stopped at the indicated timepoints and analysed under non-reducing conditions on a 4–12% SDS–PAGE gel followed by Coomassie staining.

## Preparation of 80S ribosomes and polysomes
HEK293 cells (around 80% confluency) grown in five 15 cm dishes were washed briefly with ice-cold PBS and collected in a 15 ml falcon tube. Cells were lysed in buffer containing 20 mM Tris pH 7.5, 150 mM NaCl, 5 mM $MgCl_2$, 1 mM DTT, 100 μg ml$^{-1}$ cycloheximide, 1% Triton X-100, 1× cOmplete protease inhibitor cocktail, EDTA-free (Roche) and RNasin for 10 min on ice followed by centrifugation at 13,000$g$ for 10 min. The clarified supernatant was collected and layered onto a 10–50% sucrose gradient containing 20 mM Tris pH 7.5, 150 mM NaCl, 5 mM $MgCl_2$, 1 mM DTT, 100 μg ml$^{-1}$ cycloheximide and 1% Triton X-100, followed by centrifugation at 36,000 rpm for 3 h using the SW41 Ti rotor. The fractions containing 80S ribosomes and polysomes were collected and layered onto a 50% sucrose cushion and centrifuged at 40,000 rpm for 12 h in a Type 70 Ti rotor. Ribosome pellets were then resuspended in buffer containing 20 mM HEPES pH 7.6, 100 mM KCl, 5 mM Mg(OAc)$_2$, 10 mM $NH_4Cl$ and 1 mM DTT and stored at −80 °C until further use.

## Purification of stable 60S ribosomes from HEK293 cells
60S ribosomes were purified as described previously[39,40] with minor changes. HEK293 cells were grown to around 80% confluency in fifteen 15 cm dishes with medium containing high-glucose DMEM supplemented with 10% (v/v) FBS, 50 mg ml$^{-1}$ penicillin–streptomycin and 2 mM L-glutamine. To collect cells, the medium was first removed by aspiration, washed with ice-cold PBS followed by removal of PBS by aspiration. Cells were scrapped in residual PBS and transferred to a 15 ml falcon. Cells were pelleted by centrifugation at 1,000$g$ for 3 min and the supernatant was discarded. Next, the cell pellets were resuspended in lysis buffer (containing 15 mM Tris pH 7.6, 1,500 mM NaCl, 10 mM $MgCl_2$, 1% Triton X-100, 2 mM DTT, RNAsin (60 U), 1× cOmplete mini protease inhibitor cocktail (Roche)) and mixed gently followed by incubation on ice for 10 min. The cell lysates were then centrifuged at 17,000$g$ for 10 min and the supernatant was collected. The collected supernatant was layered directly onto a high-salt sucrose cushion containing 20 mM Tris pH 7.5, 500 mM KCl, 30% (v/v) sucrose, 10 mM $MgCl_2$, 0.1 mM EDTA pH 8.0 and 2 mM DTT. Total ribosomes were sedimented by centrifugation at 63,000$g$ (24,800 rpm) for 18 h using a Type 70 Ti rotor (Beckman Coulter). The sedimented ribosomes were then resuspended in buffer containing 20 mM Tris pH 7.5, 500 mM KCl, 7.5% (v/v) sucrose, 2 mM $MgCl_2$, 75 mM $NH_4Cl$, 2 mM puromycin and 2 mM DTT. The resolubilized pellet containing ribosomes was incubated at 4 °C for 1 h and then at 37 °C for 1.5 h. To isolate 40S and 60S ribosomal subunits, the solution was layered directly onto a linear 10–30% sucrose gradient containing 20 mM Tris pH 7.5, 500 mM KCl, 6 mM $MgCl_2$ and 2 mM DTT. The 60S and 40S were separated by centrifugation at 49,123$g$ (16,800 rpm) for 9 h 42 min at 4 °C using a SW41 Ti rotor (Beckman Coulter). Gradients were fractionated into 0.5 ml fractions using the BioComp fractionating system. The fractions containing 60S ribosomal subunits were collected and exchanged into buffer containing 20 mM HEPES pH 7.2, 100 mM KCl, 5 mM $MgCl_2$, 2 mM DTT and stored at −80 °C.

## In vitro UREL–ribosome association assays
Approximately, 0.2 μM of preformed UREL was added to a mixture of 0.2 μM of 60S ribosomes (1×) and 0.5 μM of 80S ribosomes (2.5×) and incubated for 15 min at 23 °C. After incubation, the mix was layered onto a 10–50% sucrose gradient containing 20 mM HEPES pH 7.5, 50 mM KCl, 5 mM $MgCl_2$ and centrifuged at 36,000 rpm for 6 h at 4 °C. The samples were then manually fractionated into 22 fractions (100 μl

each) and analysed for co-migration by immunoblotting using the indicated antibodies.

## Cryo-EM sample preparation

**Reconstitution of stable ribosome–E3 complexes.** Approximately 10 μM of UREL complexes was incubated with 1 μM of purified 60S ribosomes in the presence of excess UFC1–UFM1 conjugate (5 μM) in buffer containing 20 mM HEPES pH 7.2, 50 mM KCl, 5 mM $MgCl_2$ and 0.25 mM TCEP for 2 h at 4 °C. After incubation, the samples were mixed with 0.05% glutaraldehyde for 30 s at 23 °C followed by quenching with 100 mM Tris pH 8.0 (final concentration). The cross-linked sample was then layered onto a 10–30% sucrose gradient containing 20 mM HEPES pH 7.5, 50 mM KCl, 5 mM $MgCl_2$ and 0.25 mM TCEP, and centrifuged using the TLS55 rotor at 24,000 rpm for 6 h at 4 °C. The sucrose gradient of 2.2 ml volume was then manually fractionated into 100 μl fractions and analysed for co-migration of UREL components, 60S ribosomes and UFC1–UFM1 by immunoblotting. The fractions containing UREL–60S ribosome–UFC1–UFM1 were then pooled and concentrated to 7.7 mg ml$^{-1}$ and buffer-exchanged to remove excess sucrose.

**Reconstitution of UFMylated 60S ribosome–UREL complexes.** First, an in vitro UFMylation reaction was performed by incubating 0.1 μM UBA5, 5 μM UFC1, 10 μM UFM1, 3 μM UFL1–UFBP1, 5 μM CDK5RAP3 and 1 μM 60S ribosomes in the presence of 5 mM $MgCl_2$ and 5 mM ATP. After the reaction, 10 μM UFC1–UFM1 was added to the reaction and further incubated at 4 °C for 2 h. The reaction products were then separated on a sucrose gradient and the fractions containing 60S–UREL–UFC1–UFM1 were collected as described in the previous section.

## Cryo-EM data collection and image processing

**UREL–60S EM grid preparation.** Cryo-grids were prepared with 0.05% glutaraldehyde-cross-linked 60S–UREL–UFC1–UFM1 complex at 7.7 mg ml$^{-1}$ in 25 mM HEPES pH 7.5, 50 mM KCl, 5 mM $MgCl_2$, 2 mM DTT. Quantifoil R3.5/1 copper 200 mesh holey grids were glow discharged using the PELCO easiGlow glow discharge unit at 15 mA for 30 s. Cryo-grids were prepared using the Thermo Fisher Scientific Vitrobot MK IV with a chamber temperature of 4 °C and 100% humidity. A total of 3 μl of protein was applied to the grid and immediately blotted for 6 s with blot force 1, followed by rapid plunge-freezing into liquid ethane.

**UREL–60S cryo-EM data collection.** Single-particle cryo-EM data were collected on the Thermo Fisher Scientific Titan Krios G2 transmission electron microscope with a Thermo Fisher Scientific Falcon 4i direct electron detector and SelectrisX energy filter. Data were collected with an accelerating voltage of 300 kV and nominal magnification of ×165,000, which corresponds to a pixel size of 0.74 Å (full data acquisition settings are shown in Extended Data Table 1). A total of 59,394 cryo-EM videos was acquired.

**UREL–60S image processing.** Cryo-EM videos were imported, beam-induced motion corrected (MOTIONCOR2) and the CTF parameters were estimated (CTFFIND4.1) using RELION (v.3.1)[41–43]. Approximately 2.2 million particles were picked from motion-corrected micrographs using crYOLO (v.1.6.1)[44] untrained particle picking (2019 general model) with a particle box size of 400 pixels and a picking confidence threshold of 0.2. Picked particles were extracted in RELION with a particle box size of 588 pixels, rescaled to 128 pixels (rescaled pixel size, ~3.4 Å). Extracted particles were imported into cryoSPARC (v.3.2)[45] for processing. Seven rounds of reference-free 2D class averages were generated with the initial classification uncertainty factor set between 2 and 7, the number of online-EM iterations set to 40 and batchsize per class set to 200, and all ribosome-like particles were taken forward. The selected 1.6 million particles were used to generate an initial 3D model with $C_1$ symmetry. The initial 3D model was further refined using the non-uniform refinement algorithm with the dynamic masking start

resolution set to a value below the resolution of the data (that is, 1 Å) to generate a refined 3D model and a mask that encompasses the entire box size. The mask and model were input for 3D variability analysis asking for three classes. Particles from the class containing ligase-bound 60S ribosomes were taken forward for another round of 3D refinement, this time with dynamic masking start resolution set to default (12 Å) and the dynamic mask threshold set to 0.1. This was then followed by several more rounds of 3D variability analysis, asking for two classes to separate ligase-bound 60S ribosomes from unbound 60S ribosomes, resulting in 356,394 ligase-bound ribosome particles. Particles were then downsampled to 128 pixels and a cryoDRGN (v.3.2.0)[46] model was trained with 8 latent dimensions and 50 training iterations. CryoDRGN particle filtering removed 57,386 junk particles, resulting in a final particle stack of 299,008 particles. The homogenous particle population containing ligase-bound ribosomes were re-extracted in Relion at the full box size. A 3D model was generated with $C_1$ symmetry, followed by non-uniform refinement with per particle defocus optimization, Ewald sphere correction and CTF refinement in cryoSPARC (v.4.2.1) to generate the ligase-bound 60S ribosome map.

To further refine the density for the ligase complex, two masks were created from the final 3D refinement volume using UCSF ChimeraX (v.1.2.5)[47]: one that encompasses the ligase complex plus RPL10a and another that encompasses the ribosome. The ribosome mask was used for particle signal subtraction. Signal-subtracted particles were then used for local refinement of the ligase complex plus RPL10a using the ligase mask to generate a ligase-only map. A cryoSPARC (v.4.2.1) 3DFlex[19] training model was generated for the ligase with 6 latent dimensions and a rigidity prior of 2. The resulting 3DFlex model was used for 3DFlex reconstruction with 40 max BFGS iterations to generate the final ligase map.

**UREL–60S model building.** The ligase-bound 60S map was sharpened using Phenix (v.1.2.1)[48] autosharpen map job and the ligase-only map was sharpened using the DeepEMhancer[49] tight target sharpening protocol. Atomic models were built using Coot (v.0.9.8.1)[50]. For the ligase-bound 60S ribosome map, PDB 7QWR (ref. 51) was used as a starting model for the 60S ribosome by rigid-body fitting the model into the density map, followed by rebuilding in Coot. No ligase components were built into the ligase-bound ribosome map except for the UFL1 PTC loop. For the ligase complex, AlphaFold2 models of the individual proteins were separated into smaller segments and then rigid-body fitted into the density map, followed by manual rebuilding in Coot. The UFL1 CTD (residues 515–786), CDK5RAP3 UUBD (residues 15–116) and UFM1 displayed poor side-chain density and the side chains of these regions were therefore set to an occupancy of 0. Atomic models were refined using Phenix real space refinement and validated using MolProbity. All 3D density maps were visually inspected in UCSF ChimeraX (v.1.2.5)[47].

**UFMylated ribosome data collection and image processing.** Cryo-EM grids were prepared as described above with 1.5 mg ml$^{-1}$ sample. Single-particle cryo-EM data were collected on the Thermo Fisher Scientific Titan Krios G2 transmission electron microscope with a Thermo Fisher Scientific Falcon 4 direct electron detector. Data were collected with an accelerating voltage of 300 kV and a nominal magnification of ×96,000, corresponding to a pixel size of 0.82 Å (full data-acquisition settings shown in Extended Data Table 1). A total of 3,028 cryo-EM videos was acquired. The data were processed as previously, with the final map being generated from the particles after several rounds of 3D variability analysis.

## XL-MS sample preparation and analysis

Approximately 1.2 μM UFL1–UFBP1, 2 μM CDK5RAP3, 0.2 μM ribosomes and 10 μM of UFC1–UFM1 were incubated with 1 mM DSBU (disuccinimidyl dibutyric urea) in buffer containing 50 mM HEPES pH 7.5, 50 mM KCl, 6 mM $MgCl_2$, 0.5 mM TCEP for 30 min at 23 °C. The reaction was

quenched by addition of 50 mM Tris pH 8.0. Cross-linked samples were processed for MS analysis using S-Trap micro spin columns (Protifi) according to the manufacturer's protocol. In brief, cross-linked samples were reduced by adding 20 mM DTT (10 min, 50 °C), and then alkylated with 40 mM iodoacetamide (30 min, 20 °C). The samples were acidified by the addition of phosphoric acid to a final concentration of 5%, and subsequently diluted with 90% methanol in 100 mM triethylammonium bicarbonate (TEAB) pH 7.1 (1:7 (v/v) sample: buffer). A total of 1 µg trypsin (Promega) was added, and the samples were then bound to a S-Trap micro spin column (Protifi). Subsequently, the column was washed three times with 90% methanol in 100 mM TEAB. An additional 0.6 µg of trypsin was applied to the column, and digestion was then performed by incubating the S-trap column at 47 °C for 90 min. Peptides were recovered by washing the column sequentially with 50 mM TEAB (40 µl), 0.2% (v/v) formic acid (40 µl) and 50% acetonitrile/0.2% (v/v) formic acid (40 µl). The eluate was then evaporated to dryness in a vacuum centrifuge and the peptides were resuspended in 5% (v/v) acetonitrile/0.1% (v/v) formic acid (20 µl) before MS analysis. Peptides (5 µl) were injected onto the Vanquish Neo LC (Thermo Fisher Scientific) system and the peptides were trapped on the PepMap Neo C18 trap cartridge (Thermo Fisher Scientific, 5 µm particle size, 300 µm × 0.5 cm) before separation using the Easy-spray reverse-phase column (Thermo Fisher Scientific, 2 µm particle size, 75 µm × 500 mm). Peptides were separated by gradient elution of 2–40% (v/v) solvent B (0.1% (v/v) formic acid in acetonitrile) in solvent A (0.1% (v/v) formic acid in water) over 80 min at 250 nl min$^{-1}$. The eluate was infused into an Orbitrap Eclipse mass spectrometer (Thermo Fisher Scientific) operating in positive-ion mode. Orbitrap calibration was performed using FlexMix solution (Thermo Fisher Scientific). Data acquisition was performed in data-dependent analysis mode and fragmentation was performed using higher-energy collisional dissociation. Each high-resolution full scan ($m/z$ 380–1,400, $R = 60,000$) was followed by high-resolution product ion scans ($R = 30,000$), with a stepped normalized collision energies of 21%, 26% and 31%. A cycle time of 3 s was used. Only charge states 3–8$^+$ were selected for fragmentation. Dynamic exclusion of 60 s was used. Cross-link identification was performed using Proteome discoverer (v.3.0) and the in-built XlinkX module (Thermo Fisher Scientific) using the following settings: crosslinker: DSBU, mass deviation tolerances of 10 ppm in MS and 0.02 Da for Sequest HT and 20 ppm for XlinkX tandem MS (MS/MS). Carbamidomethylation of Cys residues was set as a static modification, and dynamic modifications were set as Met oxidation and DSBU dead-end modifications (DSBU-amidated, DSBU Tris and DSBU hydrolysed) (maximum of three modifications per peptide). Only results with scores corresponding to a false-discovery rate of <1% were taken forward. Finally, a minimum XlinkX score of 45 was used to filter cross-linked peptides[52,53].

## Ribosome UFMylation assays

Ribosome UFMylation assays were performed as described previously[3]. Purified 60S ribosomes (approximately 0.05 µM) were mixed with 0.5 µM UBA5, 1 µM UFC1, 1 µM UFM1 and 0.1 µM UFL1–UFBP1 in a reaction buffer containing 25 mM HEPES pH 7.5, 100 mM NaCl, 10 mM MgCl$_2$ and 5 mM ATP and incubated at 37 °C for 10 min or the indicated time duration. The reaction was stopped by the addition of SDS loading buffer and run on a 4–12% SDS–PAGE gel under reducing conditions followed by immunoblotting using the indicated antibodies. In reactions containing CDK5RAP3, approximately 0.15 µM of CDK5RAP3 was added to the reaction along with 0.1 µM of UFL1–UFBP1.

## Polysome profiling using HEK293 cell lysates

Polysomes were isolated from HEK293 cells as described previously with slight modification. In brief, HEK293 cells were seeded one night before the experiment. On the day of the experiment, cells were treated with either 0.1% DMSO or 200 nM anisomycin for around 20 min before collection. Cells were washed with ice-cold PBS, scraped off

and pelleted by centrifugation at 800$g$ for 5 min. The pellet was then resuspended in lysis buffer containing 20 mM Tris pH 7.5, 150 mM NaCl, 5 mM MgCl$_2$, 1 mM DTT, 100 µg ml$^{-1}$ cycloheximide, 0.02% Digitonin, cOmplete protease inhibitor cocktail, EDTA-free (Roche) and RNasin. Digitonin-treated cells were incubated for 5 min on ice and centrifuged at 17,000$g$ for 10 min at 4 °C. The supernatant containing the cytoplasmic extract was discarded and the remaining pellet was washed with 20 mM Tris pH 7.5, 150 mM NaCl and 5 mM MgCl$_2$ and centrifuged at around 7,000$g$ for 5 min. Supernatant was discarded and the pellet was resuspended in lysis buffer containing 20 mM Tris pH 7.5, 150 mM NaCl, 5 mM MgCl$_2$, 1 mM DTT, 100 µg ml$^{-1}$ cycloheximide, 0.5% Triton X-100, cOmplete protease inhibitor cocktail, EDTA-free (Roche) and RNasin. The resuspended pellets were incubated on ice for 10 min and centrifuged at 17,000$g$ for 10 min at 4 °C. The supernatant containing the membrane fraction extract was transferred to a new Eppendorf tube and the amount of RNA was quantified for each sample using the NanoDrop system. RNA-normalized samples were then layered onto a 10–50% sucrose gradient containing 20 mM Tris pH 7.5, 150 mM NaCl, 5 mM MgCl$_2$, 1 mM DTT, 100 µg ml$^{-1}$ cycloheximide and RNAsin, and then centrifuged at 36,000 rpm for 3 h. Polysomes were then separated by fractionation using the Biocomp fractionating system and analysed using western blotting.

## Comparison of 60S and 80S UFMylation in vitro

In vitro UFMylation reactions were performed by incubating 0.1 µM 60S ribosome, 0.2 µM or 0.3 µM enriched 80S (two or threefold excess over 60S) with 0.5 µM UBA5, 1 µM UFC1, 1 µM UFM1, 0.3 µM UFL1–UFBP1 and 0.3 µM CDK5RAP3 in the presence of 5 mM MgCl$_2$ and 5 mM ATP at 37 °C for 15 min. After incubation, the reaction mix was layered over a 10–50% sucrose gradient containing 20 mM Tris pH 7.5, 150 mM NaCl, 5 mM MgCl$_2$, 1 mM DTT and centrifuged at 36,000 rpm for 3 h at 4 °C using a SW41 Ti rotor. The gradients were fractionated using the Bio-Comp fractionation system. The sucrose gradient fractions were then run on a 4–12% SDS–PAGE gel and analysed for UFMylation of RPL26 by immunoblotting.

## Preparation of membrane-associated 60S ribosomes

Parental cells (WT HEK293) or *CDK5RAP3*-KO cells (around 80% confluency) grown in ten 15 cm dishes were washed briefly with ice-cold PBS and collected in a 15 ml falcon tube. Cells were pelleted down by centrifugation at 500$g$ for 5 min. Cell pellets were resuspended in buffer containing 20 mM Tris pH 7.5, 150 mM NaCl, 5 mM MgCl$_2$, 1 mM DTT, 100 µg ml$^{-1}$ cycloheximide, 0.02% (w/v) digitonin, 1× cOmplete protease inhibitor cocktail, EDTA-free (Roche) and RNasin for 10 min on ice followed by centrifugation at 17,000$g$ for 10 min. The clarified supernatant is the cytosolic fraction and was discarded. The remaining membrane pellet was resuspended in lysis buffer containing 20 mM Tris pH 7.5, 150 mM NaCl, 5 mM MgCl$_2$, 1 mM DTT, 100 µg ml$^{-1}$ cycloheximide, 1% (w/v) decyl maltose neopentyl glycol (DMNG), 1× cOmplete protease inhibitor cocktail, EDTA-free (Roche) and RNasin for 15 min on ice, and then centrifuged at 17,000$g$ for 10 min. The clarified supernatant was collected and layered onto a 10–30% sucrose gradient containing 20 mM Tris pH 7.5, 150 mM NaCl, 5 mM MgCl$_2$, 1 mM DTT, 100 µg ml$^{-1}$ cycloheximide and 0.01% DMNG, and then centrifuged at 36,000 rpm for 3 h using the SW41 Ti rotor. Fractions containing 60S ribosomes were collected and exchanged into buffer containing 20 mM HEPES pH 7.2, 100 mM KCl, 5 mM MgCl$_2$ and 2 mM DTT and stored at −80 °C until use.

## In vitro 60S ribosome–SEC61 dissociation assays

The in vitro 60S–SEC61 dissociation reaction was performed by incubating 0.05 µM membrane solubilized 60S ribosomes (60S–SEC61 solubilized and enriched from *CDK5RAP3*-KO cells) with 0.5 µM UBA5, 1 µM UFC1, 1 µM UFM1, 0.1 µM UFL1–UFBP1, 0.1 µM CDK5RAP3 in the presence of 5 mM MgCl$_2$ and 5 mM ATP at 37 °C for 25 min. At the end

of the reaction, the reaction mix was layered over a 10–50% sucrose gradient containing 20 mM Tris pH 7.5, 150 mM NaCl, 5 mM $MgCl_2$ and 1 mM DTT, and centrifuged at 36,000 rpm for 3 h using the SW41 Ti rotor. Sucrose gradients were fractionated using the BioComp fractionation system. The sucrose gradient fractions were separated on a 4–12% SDS–PAGE gel and analysed for co-migration of SEC61β with 60S ribosomes by immunoblotting.

### LC–MS/MS sample preparation, data acquisition and analysis
First, an in vitro ribosome UFMylation reaction was performed to generate UFMylated ribosomes in the presence of either UFL1–UFBP1 or UREL. Then, the reaction products were run on a 4–12% SDS–PAGE gel to separate the mono- and di-UFMylated ribosomes. Next, the bands corresponding to mono- and di-UFMylated ribosomes were excised and in-gel digestion was performed according to a previously described protocol[54]. Digested peptides were analysed by liquid chromatography coupled with MS/MS (LC–MS/MS) on the Exploris 240 (Thermo Fisher Scientific) system coupled to the Evosep One (Evosep). The samples were loaded onto the Evotips according to the manufacturer's recommendations and analysed using the 30 SPD method. Peptides were then analysed in on the Exploris 240 system using data-dependant acquisition with an MS1 resolution of 60,000, an AGC target of 300% and a maximum injection time of 25 ms. Peptides were then fragmented using TOP 2 s method, MS2 resolution of 15,000, NCE of 30%, AGC of 100% and maximum injection time of 100 ms. Peptide identification was performed in MaxQuant (v.2.0.2.0) against UniProt SwissProt Human containing isoforms (released 5 May 2021) with match between runs enabled. Carbamidomethylation (C) was set as a fixed modification and oxidation (M), acetyl (protein N-term) and the addition of the dipeptide valine–glycine (K) were set as variable modifications. The other parameters were left as the default.

### Preparation of isopeptide-linked UFC1–UFM1 conjugate
First, 30 μM UBA5, 30 μM UFC1(C116K) and 60 μM UFM1 were incubated in 25 mM HEPES pH 7.5, 200 mM NaCl, 10 mM $MgCl_2$ and 10 mM ATP. The pH of the reaction mixture was adjusted to 9.8 with 0.5 M CAPS, pH 11.5, and incubated for 18 h at 23 °C. UFC1–UFM1 was subsequently separated from UBA5 and unreacted UFC1 and UFM1 using the HiLoad 26/600 Superdex 75 pg column, pre-equilibrated with 25 mM HEPES pH 7.5, 200 mM NaCl, 1 mM DTT.

### Crystallization and structure determination
**UFC1–UFM1 conjugate.** UFC1–UFM1 crystals were obtained using the sitting-drop vapour diffusion technique whereby UFC1–UFM1 (22.8 mg ml$^{-1}$) was 1:1 mixed with 30% (v/v) PEG 400, 0.1 M Tris pH 8.5, 0.2 M Na citrate and incubated at 19 °C. Single crystals appeared within 2–3 days. Crystals were flash-frozen in crystallization buffer containing 30% (v/v) ethylene glycol. Datasets were collected at Diamond Light Source (DLS), beamline I04, and processed with Xia2 (ref. 55) and DIALS[56]. The crystal structure was solved by molecular replacement (PHASER)[57] using the crystal structures of UFC1 (PDB: 3EVX)[58] and UFM1 (PDB: 5IA7)[59] as the starting model. Refinement and model building was performed using REFMAC[60] and Coot[50] (CCP4i2 suite), respectively. The statistics for data collection and refinement are listed in Extended Data Table 2.

**UFL1–UFBP1–UFC1 complex.** UFL1–UFBP1–UFC1 crystals were obtained using the sitting-drop vapour diffusion technique whereby UFL1–UFC1–UFBP1 (20.2 mg ml$^{-1}$) was 1:1 mixed with 1.03 M $Li_2SO_4$, 0.1 M HEPES pH 7.2 and incubated at 19 °C. Single crystals appeared within 1–2 days. Crystals were flash-frozen in crystallization buffer containing 30% (v/v) ethylene glycol. Datasets were collected at the European Synchrotron Radiation Facility (ESRF), beamline ID23-EH2, and processed with the autoPROC suite[61] (including XDS[62], Pointless[63] Aimless[64], CCP4 (ref. 65) and STARANISO[66]). The crystal structure was solved by molecular

replacement (PHASER)[57] using the AlphaFold[11] predicted models for UFL1 and UFBP1 and the crystal structure of UFC1 (PDB: 3EVX)[58] as starting models. Refinement and model building was performed using REFMAC[60] and Coot[50] (CCP4i2 suite), respectively. The statistics for data collection and refinement are listed in Extended Data Table 2.

**SEC.** Analytical SEC runs were performed using the Superdex 200 Increase 3.2/300 column, pre-equilibrated with 25 mM HEPES pH 7.5, 200 mM NaCl, 0.5 mM TCEP. A total of 50 μl protein of the different components was mixed and incubated on ice for 30 min before loading onto the column.

**ITC.** ITC experiments were performed using a MicroCal PEAQ-ITC (Malvern). Proteins were first dialysed into ITC buffer containing 25 mM HEPES pH 7.5, 200 mM NaCl, 0.44 mM TCEP. Each experiment consisted of 13 injections for a duration of 6 s each followed by a 150 s spacing between injections except the experiment for UFBP1 UFIM–UFM1, which consisted of 19 injections instead. All of the experiments were performed at 25 °C.

**Figures.** Adobe Illustrator, BioRender and ChimeraX[47] were used to make figures.

**Materials availability.** All cDNA constructs in this study were generated by H.M.M., J.J.P. and the cloning team at the MRC PPU Reagents and Services team. All of the plasmids have been deposited at the MRC PPU Reagents and Services and are available at https://mrcppureagents.dundee.ac.uk/.

### Reporting summary
Further information on research design is available in the Nature Portfolio Reporting Summary linked to this article.

## Data availability
The cryo-EM coordinates have been deposited at the PDB under accession codes 8QFD (ligase-bound 60S) and 8QFC (ligase only). Cryo-EM maps have been deposited at the Electron Microscopy Data Bank under accession codes EMD-18382 (ligase-bound 60S) and EMD-18381 (ligase only). X-ray structure factors and associated models have been deposited at the PDB under accession codes 8C0D and 8BZR. The raw DSBU XL-MS data and the LC–MS/MS analysis of ribosome UFMylation have been deposited at the ProteomeXchange Consortium via the PRIDE[67] partner repository under dataset identifiers PXD046990 and PXD046991, respectively. Source data are provided with this paper.

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

**Acknowledgements** We thank J. Aspden, J. Fontana, R. Hay and the members of the Zeqiraj and Kulathu laboratories, especially S. Matthews, L. Krshnan and A. Perez for discussions and reading the manuscript; R. Hegde for the gift of SEC61-β antibodies; and the staff at Leeds Cryo-EM facility, Diamond beamline I04 and ESRF beamline ID23-EH2 for assistance. This work was supported by a Wellcome Trust Senior Fellowship 222531/Z/21/Z (to E.Z.), Wellcome Trust Four-Year PhD Studentship in Basic Science 222372/Z/21/Z (to L.M.), Wellcome & Royal Society Sir Henry Dale Fellowship 220628/Z/20/Z (to A.N.C.), University Academic Fellowship from the University of Leeds (to A.N.C.). BBSRC BB/T008172/1 (to Y.K.), ERC Starting grant RELYUBL, 677623 (to Y.K.), MRC grant MC_UU_00018/3 (to Y.K.) and the Lister Institute of Preventive Medicine (to Y.K.). Wellcome (223810/Z/21/Z) funded the MS equipment. The Astbury cryo-EM Facility is funded by a University of Leeds ABSL award and Wellcome Trust grant (221524/Z/20/Z), the University of Dundee CryoEM facility is funded by Wellcome Trust (223816/Z/21/Z), MRC (MRC World Class Laboratories PO4050845509).

**Author contributions** L.M., J.J.P., H.M.M. and R.T. designed, performed and interpreted most of the experiments with input from Y.K. and E.Z. L.M. performed cryo-EM experiments, data processing and model building. J.J.P. prepared UREL–60S complexes, UREL–UFMylated 60S complexes for EM analysis, prepared samples for XL-MS and MS UFMylation site mapping, prepared polysomes and 80S ribosomes and biochemical assays. H.M.M. performed X-ray crystallography experiments, biochemical and enzymatic assays, model building, analytical SEC together with G.H. and ITC measurements. H.M.M. and J.J.P. performed all of the cloning and purification of all of the proteins used in this work. R.T. performed all cell-based experiments, polysome profiling, ribosome UFMylation assays and experiments to show SEC61–60S association and splitting. M.F. performed initial EM and biochemical/biophysical analyses of ligase complexes provided by J.J.P. T.M. designed CRISPR gRNAs. D.M. generated *CDK5RAP3*-KO cells. T.C.M. and A.N.C. performed XL-MS experiments. J.V. and F.L. performed MS analyses to identify RPL26 UFMylation sites. Y.K. and E.Z. supervised the project and wrote the first draft of the manuscript with input from L.M., J.J.P. and H.M.M. All of the authors critically reviewed the manuscript.

**Competing interests** The authors declare no competing interests

**Additional information**
**Correspondence and requests for materials** should be addressed to Elton Zeqiraj or Yogesh Kulathu.

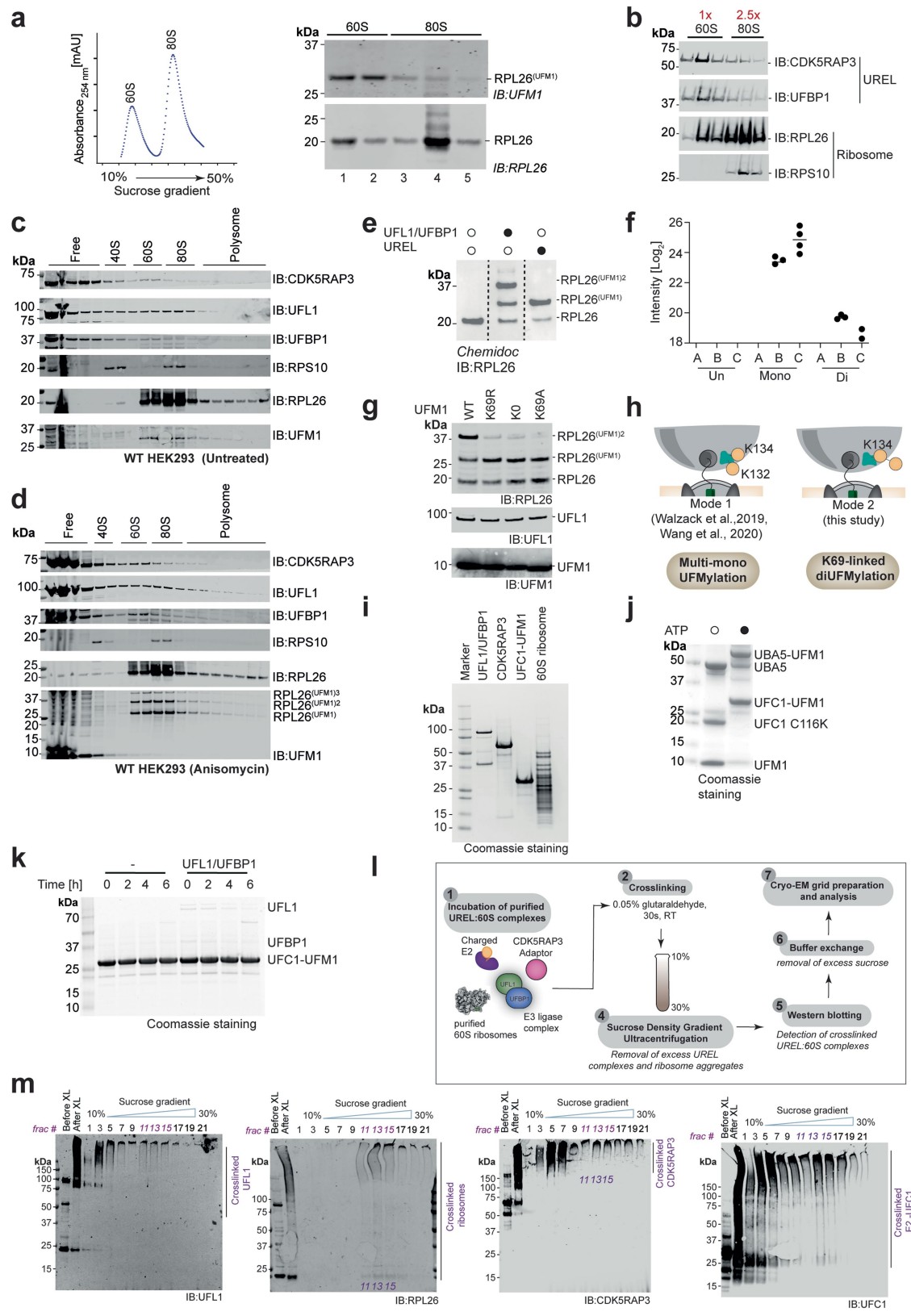

**Extended Data Fig. 1** | See next page for caption.

**Extended Data Fig. 1 | Preparation of complexes for cryo-EM. a**, UREL selectively UFMylates 60S *in vitro*. Ribosome UFMylation by the addition of UBA5, UFC1, UFM1 and UREL (UFL1, UFBP1 and CDK5RAP3) to a mixture of 60S ribosomes and 2-fold excess of 80S ribosomes. The reaction mixture was separated on a sucrose density gradient (left) and the fractions analysed for RPL26 UFMylation by immunoblotting (right). Data are representative of n = 3 independent experiments. **b**, UREL binds to 60S ribosomes. UREL was incubated with a mixture of 60S and 2.5-fold excess 80S ribosomes, separated on a sucrose density gradient and analysed by immunoblotting with the indicated antibodies. **c & d**, UREL associates with UFMylated 60S ribosomes in cells. Membrane fractions from HEK 293 WT cells left untreated (top) or treated with anisomycin (bottom) were separated on a sucrose density gradient and the different fractions were immunoblotted using the indicated antibodies. **e**, *In vitro* UFMylation assay to generate UFMylated 60S ribosomes in the presence of UFL1/UFBP1 or UREL for LC-MS/MS analysis. **f**, The reaction products from (**e**) were analysed by LC-MS/MS to identify the UFMylation site, linkage type and to quantify UFMylation of RPL26 under different conditions. Abundance of K134-GG remnants in the presence of E1 and E2 alone (A), UFL1/UFBP1 (B) and UREL (C). (Un: Unmodified RPL26, Mono: MonoUFMylated RPL26, Di: DiUFMylated RPL26) (n > 2 technical replicates). **g**, Immunoblot showing UFMylation of RPL26 in the presence UFM1 WT, K69R, K69A or lysine-less UFM1(K0). **h**, Mode of UFMylation of RPL26 as inferred from LC-MS/MS and biochemical experiments from **e** to **g**. **i**, Coomassie stained SDS-PAGE gel of purified UREL, UFC1-UFM1 and 60S ribosomes used in the preparation of samples for visualization by cryo-EM. **j**, Coomassie stained SDS-PAGE gel showing *in vitro* reaction for the generation of UFC1-UFM1. **k**, Coomassie stained SDS-PAGE gel analysing stability of UFC1-UFM1. Purified UFC1-UFM1 was incubated with UFL1/UFBP1 at 37 °C to monitor hydrolysis of UFC1-UFM1 conjugate. The reaction was stopped at indicated time points and separated on a 4-12% SDS-PAGE gel followed by Coomassie staining. **l**, Schematic showing reconstitution of stable UREL:60S complexes for cryo-EM analysis. **m**, Preparation of stable UREL:60S complex as outlined in **l**.

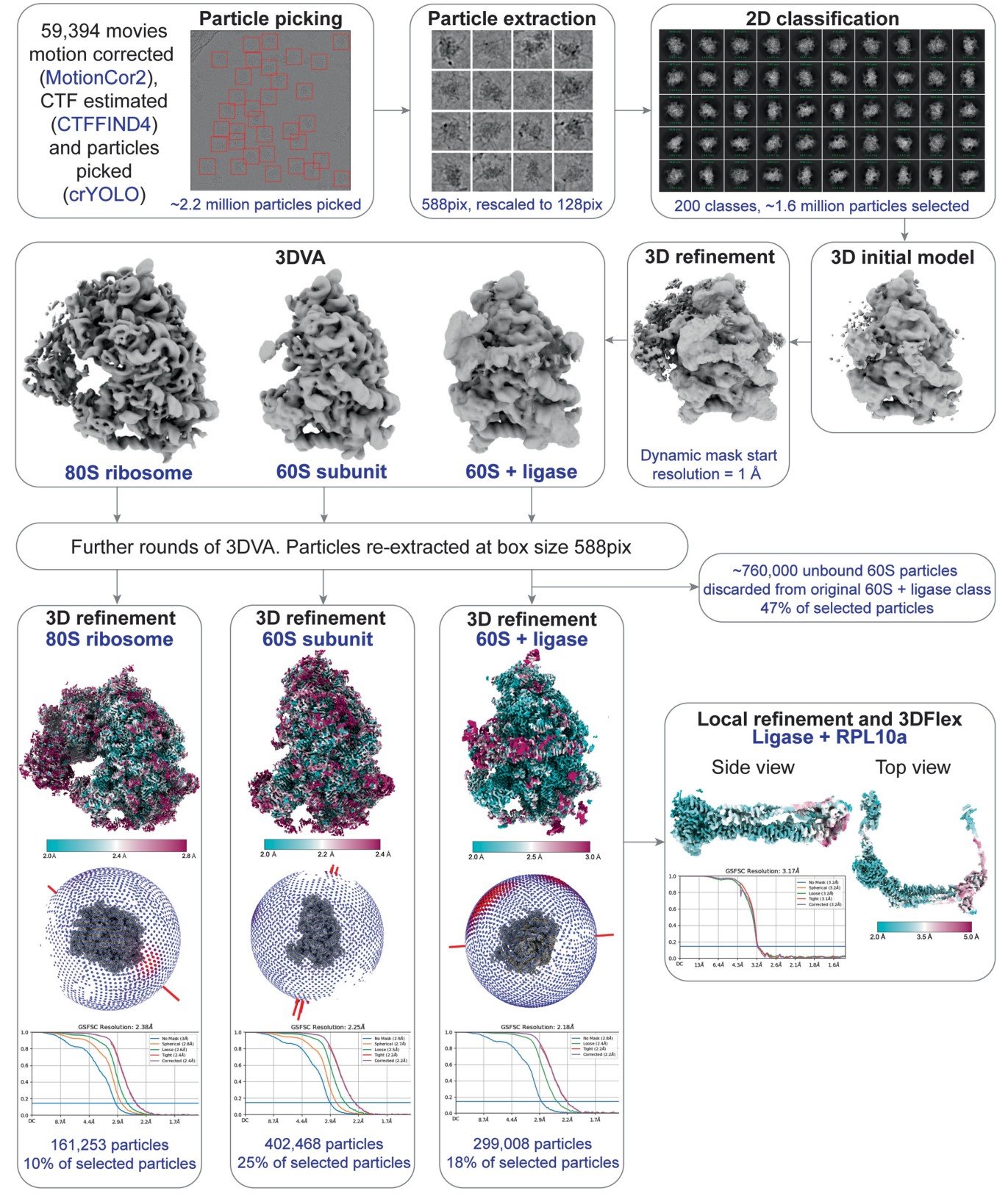

**Extended Data Fig. 2** | See next page for caption.

**Extended Data Fig. 2 | Cryo-EM data processing pipeline.** Cryo-EM data processing steps to obtain cryo-EM maps for the 80S ribosome, 60S ribosome subunit, UREL ligase-bound 60S ribosome and UREL bound to RPL10a. -2.2 million picked particles were extracted using a box size of 588 pixels (pix), rescaled to 128 pix. After several rounds of 2D classification -1.6 million ribosome-like particles were selected. All ribosome-like particles were pooled to generate an initial 3D model, followed by 3D refinement. 3D variability analysis (3DVA) separated three major classes: 80S ribosome, 60S subunit and UREL-bound 60S (60S+ligase). These underwent further rounds of 3DVA and cryoDRGN particle sorting to obtain homogenous particles, which were then re-extracted using the original box size, followed by a final 3D refinement. To generate the ligase+RPL10a map, signal corresponding to the 60S ribosome was subtracted and the region corresponding to the UREL ligase and RPL10a was locally refined. This was then further refined using 3DFlex training and reconstruction. Final maps are coloured by local resolution. 3D angular distribution representation and FSC curves are shown, calculated using the gold standard FSC cutoff of 0.143.

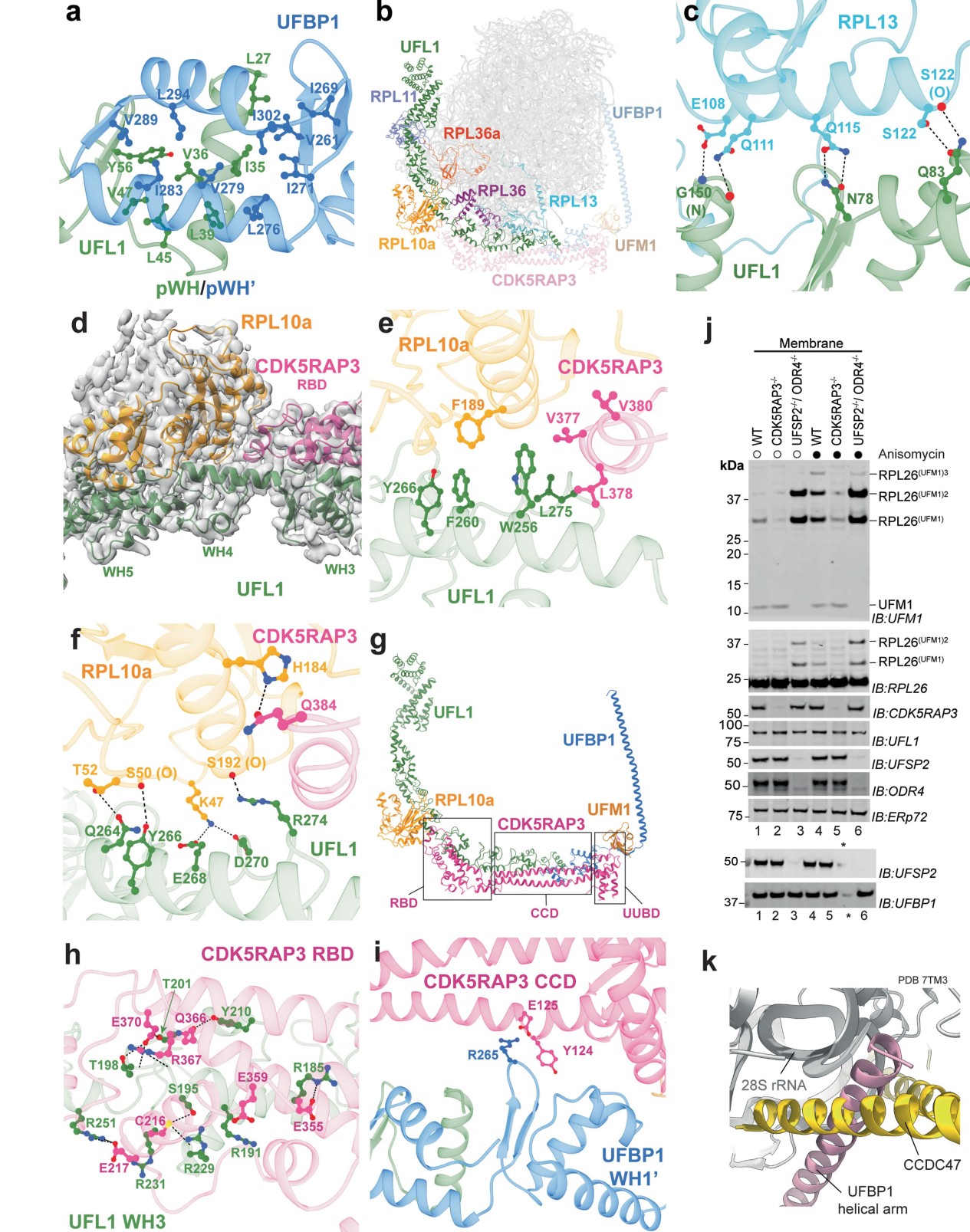

**Extended Data Fig. 3** | See next page for caption.

**Extended Data Fig. 3 | UREL:60S subunit interactions.** Main interactions between UREL and the 60S ribosome. Throughout, side chains are displayed as ball and stick and hydrogen bonds shown as black dashed lines. **a**, UFL1 N-terminus and UFBP1 C-terminus form a composite winged helix domain (pWH/pWH'). **b**, 60S ribosomal proteins RPL10a, RPL11, RPL36a, RPL36 and RPL13 interact with UFL1. **c**, Hydrogen bond network between UFL1 winged helix domains WH1 and WH2 and RPL13. **d**, UFL1 and CDK5RAP3 bind to RPL10a of the L1 stalk. Atomic model cartoon is coloured by protein and cryo-EM density shown in transparent grey. RBD is CDK5RAP3 ribosome binding domain. **e**, Hydrophobic residues at the RPL10a:UREL interface. **f**, Hydrogen bonding residues at the RPL10a:UREL interface. **g**, Overview of CDK5RAP3 domains. RBD is RPL10a binding domain. CCD is coiled-coil domain. UUBD is UFM1/UFBP1 binding domain. **h**, Main electrostatic interactions between CDK5RAP3 RBD and UFL1. **I**, Main electrostatic interactions between CDK5RAP3 CCD and UFBP1. **j**, Immunoblotting of membrane fractions from HEK293 WT, CDK5RAP3 KO or UFSP2, ODR4 double KO cells untreated or treated with 200 nM anisomycin for 60 min. Asterisk indicates empty lane. **k**, Superposition of CCDC47 (PDB ID 7tm3) with cryoEM structure of UREL:60S complex shown in cartoon representation reveals similar mode of ribosome docking.

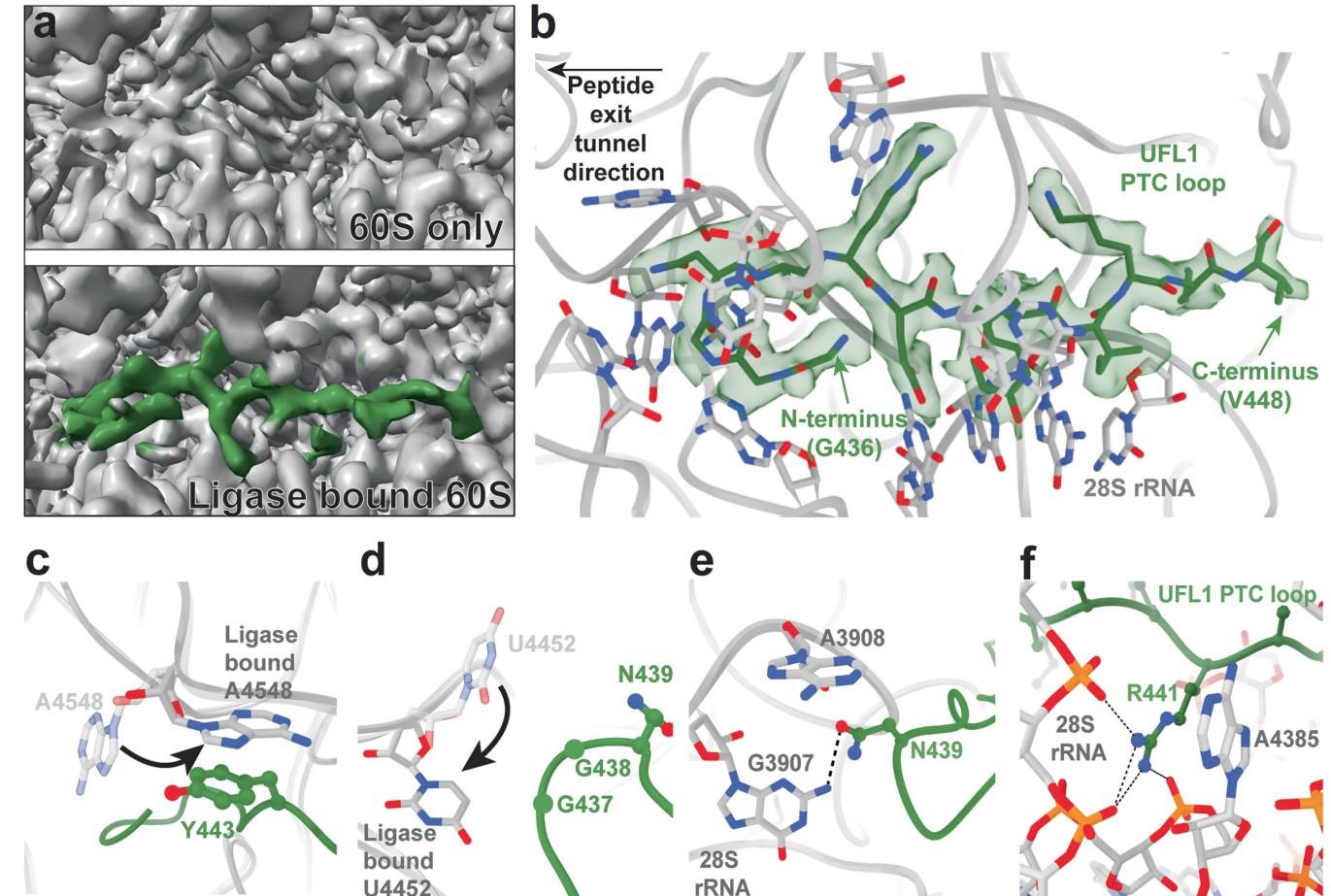

**Extended Data Fig. 4 | UFL1 loop binds near the P-site of the peptidyl transferase centre (PTC). a**, Comparison between P-site regions of the 60S only map versus the UREL-bound 60S map, viewed at similar thresholds. Additional density was observed in the ligase-bound 60S map which corresponds to UFL1 (green). **b**, UFL1 loop positioning within PTC with surrounding rRNA bases shown. Cryo-EM density for UFL1 loop shown in transparent green. Arrows indicate loop N- and C-termini. **c**, 28S A4548 moves towards P-site to stack with UFL1 Y443. Transparent grey 60S model represents non-ligase bound 60S (PDB ID 6r7q). Opaque grey 60S model is UREL bound 60S. **d**, 28S U4452 moves towards A-site to sit proximal to G437. Transparent grey 60S model represents non-ligase bound 60S (PDB ID 6r7q). Opaque grey 60S model is UREL bound 60S. **e**, UFL1 N439 stacks with 28S rRNA A3908 and hydrogen bonds with G3807 (dashed line). **f**, UFL1 R441 stacks with 28S rRNA A4385 and hydrogen bonds with surrounding phosphates of 28S rRNA (dashed lines).

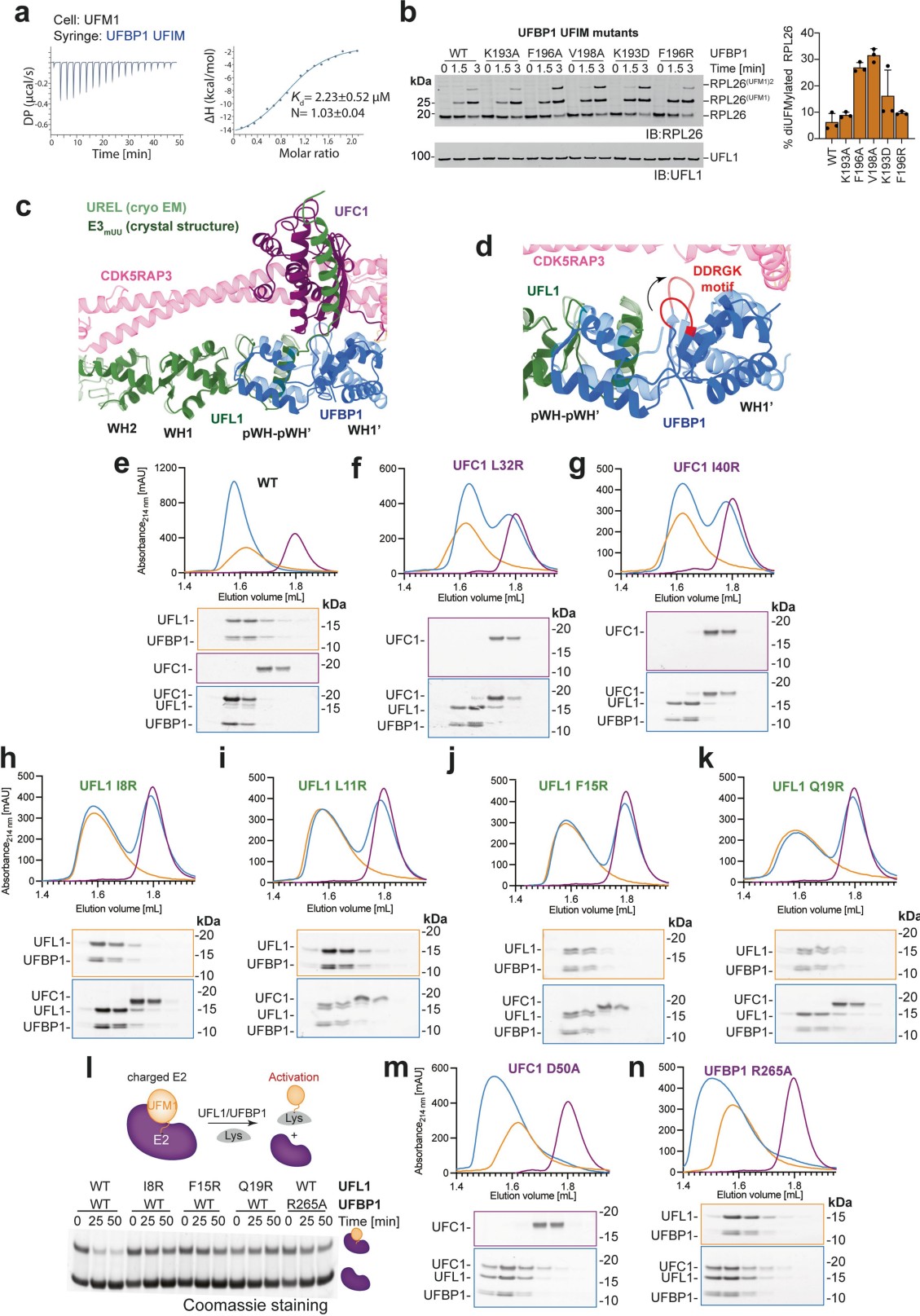

**Extended Data Fig. 5** | See next page for caption.

**Extended Data Fig. 5 | Analysis of UFBP1 UFIM:UFM1 and UREL:UFC1-UFM1 interactions. a**, ITC titration curve and the corresponding fitting curve for UFM1 and UFBP1 UFIM (178-204). Data are representative of n = 2 independent experiments. **b**, (Left) *In vitro* UFMylation assays with UFBP1 UFIM mutants and immunoblotted with the indicated antibodies. Data are representative of n = 3 independent experiments. (Right) Quantitative representation showing percentage of di-UFMylation (mean ± SD; n = 3) in the *in vitro* UFMylation assays. **c**, Overlay of the cryo-EM structure of the UREL complex and the crystal structure of $E3_{mUU}$ in complex with UFC1. **d**, Close-up view showing the conformational change of the DDRGK motif of UFBP1, highlighted in red, in the cryo-EM structure. **e**, Gel filtration chromatograms of $E3_{mUU}$-ΔUFIM:UFC1 complex. Approximately 40 μM $E3_{mUU}$ were incubated with 15 μM UFC1 for 15 min at 4 °C and loaded on a Superdex 200 3.2/300 gl column. The corresponding peak fractions were collected and analysed on a 4-12% SDS-PAGE gel followed by Coomassie staining. **f**, to **k**, Gel filtration chromatograms of $E3_{mUU}$-ΔUFIM:UFC1 complexes with mutations at the UFL1 α1/UFC1 α2 interface. **l**, Lysine discharge assays in the presence of $E3_{mUU}$ mutants. (Top) Schematic describing the assay workflow. (Bottom) Coomassie stained SDS-PAGE gel showing aminolysis of UFM1 from UFC1-UFM1 in the presence of $E3_{mUU}$-ΔUFIM mutants. **m**, and **n**, SEC elution profiles of $E3_{mUU}$-ΔUFIM:UFC1 complexes with mutations disrupting the UFBP1 R265:UFC1 D50 interaction.

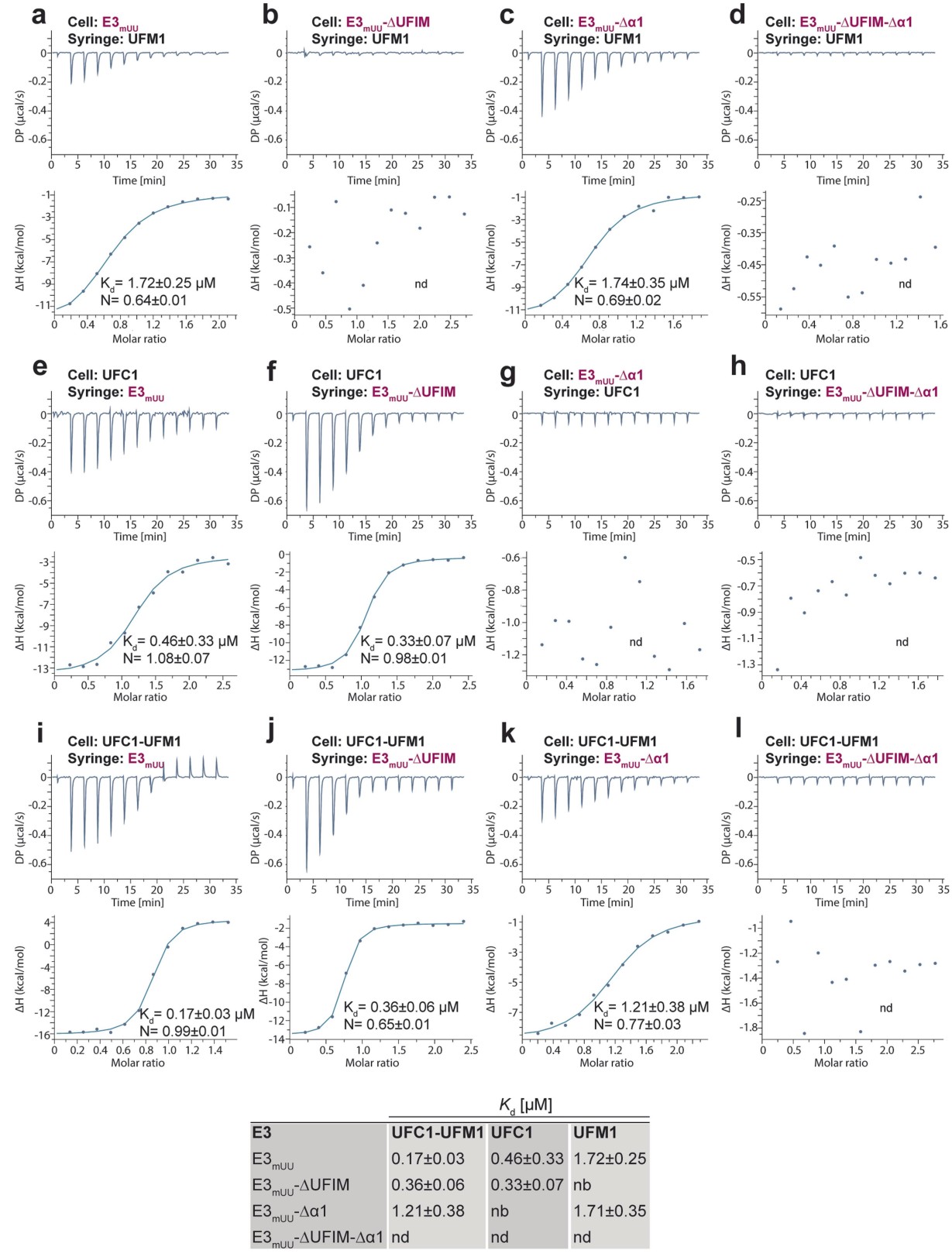

**Extended Data Fig. 6 | Binding curves for ITC experiments. a–l,** Representative ITC binding curves analysing interactions between indicated E3$_{mUU}$ proteins and UFC1, UFM1 or UFC1-UFM1. All experiments were performed in duplicates, the dissociation constants and stoichiometries were calculated based on both experiments. (Bottom) Summary of disassociation constants of the different E3$_{mUU}$ constructs with UFC1-UFM1, UFC1 and UFM1 measured by ITC. nd indicates that no binding was detected.

| E3 | $K_d$ [µM] | | |
|---|---|---|---|
| | UFC1-UFM1 | UFC1 | UFM1 |
| E3$_{mUU}$ | 0.17±0.03 | 0.46±0.33 | 1.72±0.25 |
| E3$_{mUU}$-ΔUFIM | 0.36±0.06 | 0.33±0.07 | nb |
| E3$_{mUU}$-Δα1 | 1.21±0.38 | nb | 1.71±0.35 |
| E3$_{mUU}$-ΔUFIM-Δα1 | nd | nd | nd |

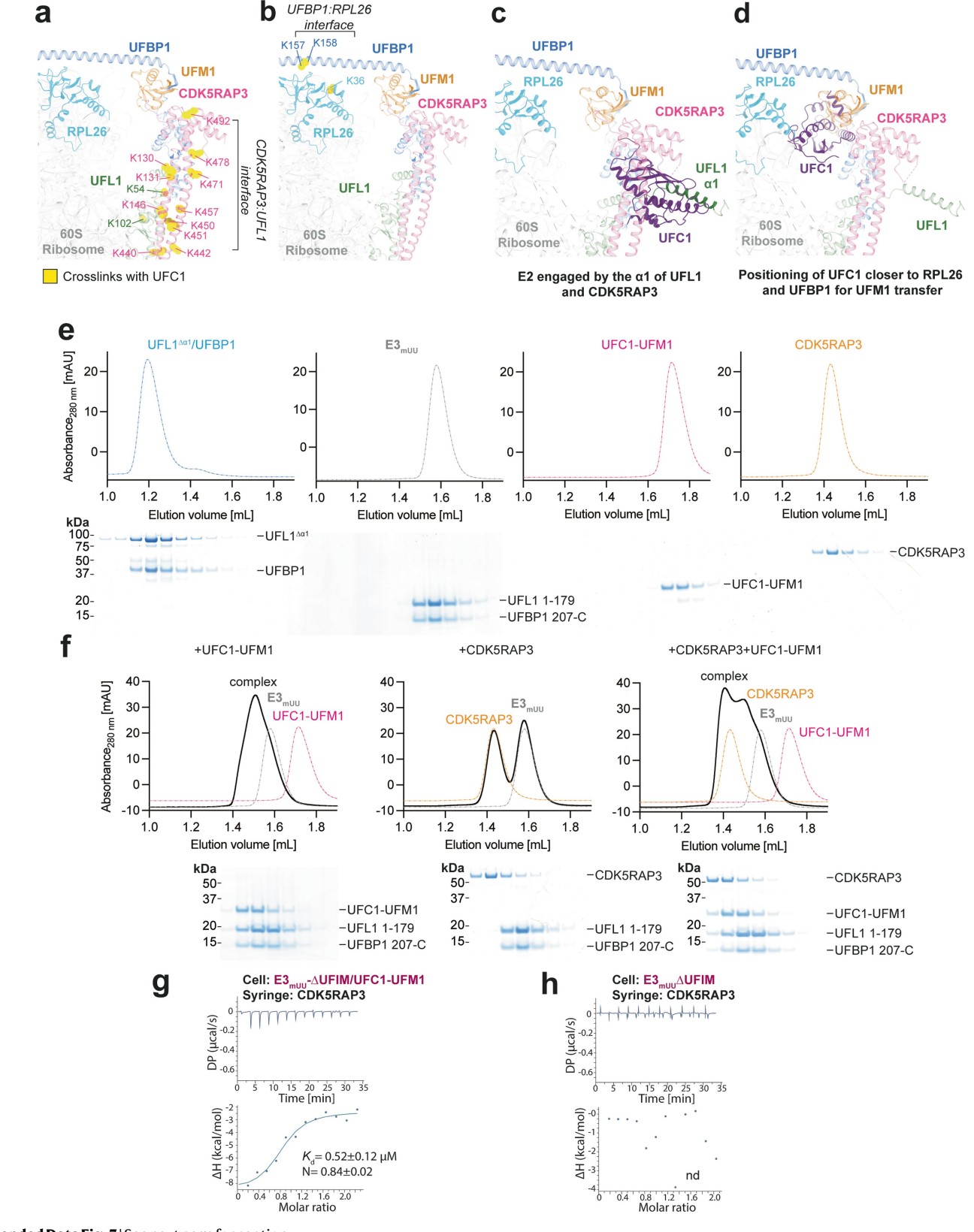

**Extended Data Fig. 7 |** See next page for caption.

**Extended Data Fig. 7 | Existence of a composite binding site for UFC1-UFM1 on UREL. a**, Mapping crosslinked residues on the UREL:60S cryo-EM structure. Residues on CDK5RAP3 and UFL1 crosslinked with UFC1 are highlighted in yellow. **b**, Residues on UFBP1 and RPL26 crosslinked with UFC1 may constitute two distinct interfaces for charged E2 interaction. Cryo-EM model of UREL:60S is shown in cartoon representation and crosslinked residues shown as ball and sticks are highlighted in yellow. (**c-f**) Models depicting different intermediate stages of UFC1 prior to conjugation of UFM1 on RPL26. **c**, Model to show that UFC1 is potentially engaged closer to the interface formed by CDK5RAP3 and UFL1 as observed in XL-MS data shown in (**a**). Model was generated by superposition of crystal structures of $E3_{mUU}$ bound to UFC1 and cryo-EM structure of UREL:60S ribosome. **d**, Model generated by superposition of crystal structure of UFC1-UFM1 conjugate onto cryo-EM structure of UREL:60S ribosome to suggest UFC1's proximity to RPL26 and UFBP1 as observed in the XL-MS data shown in (**b**). **e**, (Top) Individual SEC elution profiles for UFL1-$\Delta\alpha1$/UFBP1, $E3_{mUU}$, CDK5RAP3 and UFC1-UFM1. (Bottom) The corresponding peak fractions were separated on a 4-12% SDS-PAGE gel under reducing conditions and Coomassie stained. **f**, (Top) Gel filtration chromatograms of $E3_{mUU}$:UFC1-UFM1, $E3_{mUU}$:CDK5RAP3 and $E3_{mUU}$:CDK5RAP3/UFC1-UFM1. (Bottom) The corresponding peak fractions were separated on a 4-12% SDS-PAGE gel under reducing conditions and Coomassie stained. **g**, ITC binding curves for preformed $E3_{mUU}$/UFC1-UFM1 complex and CDK5RAP3. Data are representative of n = 2 independent experiments. The dissociation constant and stoichiometry were calculated based on both experiments. **h**, Control experiment for (**g**), where no UFC1-UFM1 was added to $E3_{mUU}$. nd indicates that no binding was detected. Data are representative of n = 2 independent experiments.

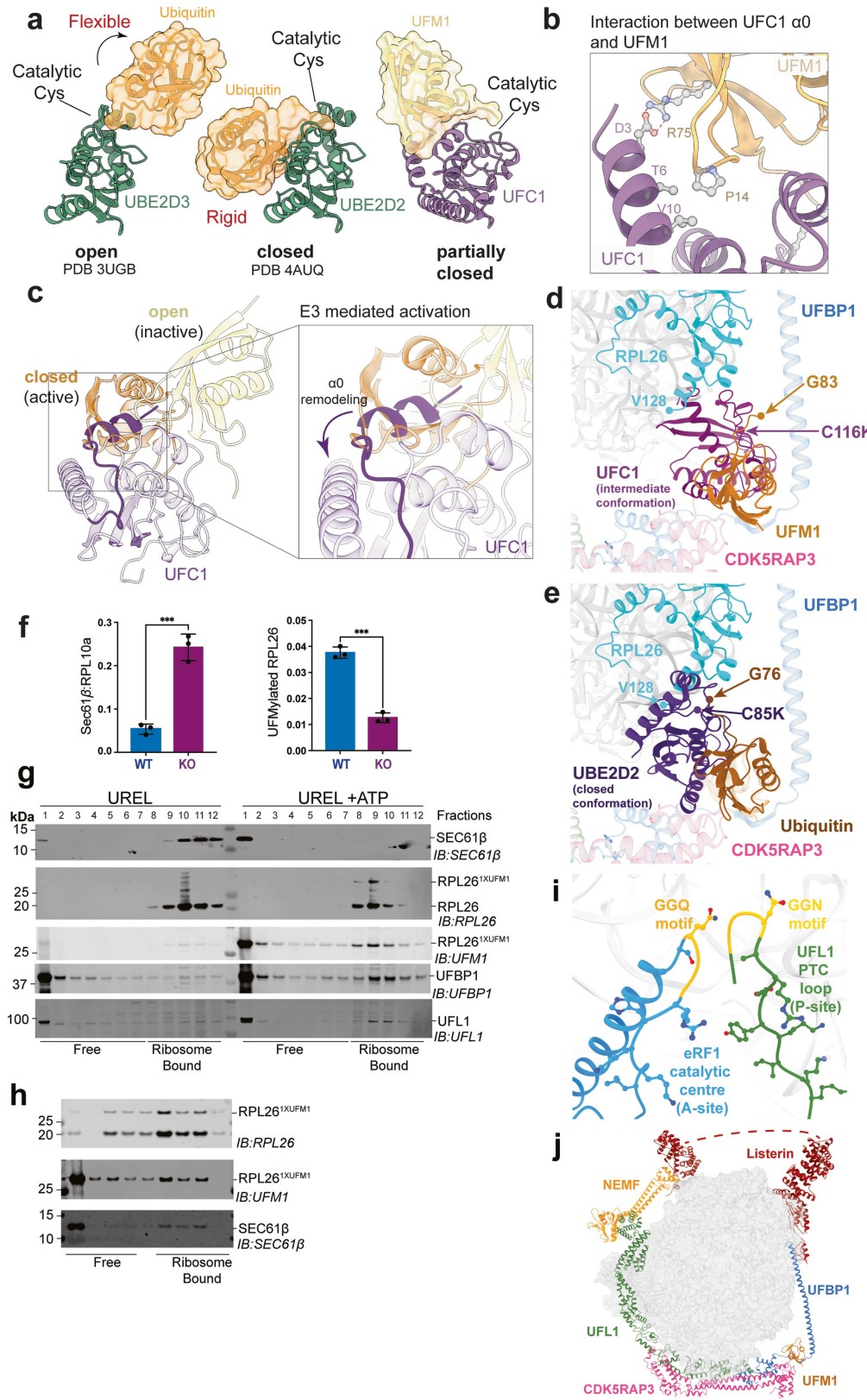

**Extended Data Fig. 8** | See next page for caption.

**Extended Data Fig. 8 | Active conformation of UFC1–UFM1 and UFMylation-dependent SEC61 dissociation from 60S. a**, Comparison of apo and E3 bound states of E2-Ubiquitin conjugate (PDB IDs 3ugb, 4auq) with apo UFC1-UFM1 conjugate. Ubiquitin (dark orange) and UFM1 (light orange) are shown as surfaces overlaid on cartoons. UBE2D2/3 (green) and UFC1 (purple) are shown in cartoon representation. **b**, Enlarged view of the interface between UFC1 α0 and UFM1 with the interacting residues highlighted in ball and stick representation. **c**, Model depicting open and closed states of UFC1-UFM1 conjugate. Inset highlights the clash between UFC1 α0 and an incoming UFM1 suggesting a requirement to remodel UFC1 α0 to accommodate UFM1 in the closed-active state. **d**, Crystal structure of UFC1-UFM1 conjugate in an intermediate conformation superimposed onto cryo-EM structure of UREL:UFM1 bound 60S ribosome. C-terminal glycine of UFM1 (G83), catalytic cysteine to lysine mutation of UFC1 (C116K) and most C-terminal residue of RPL26 for which density is present in the cryo-EM map (V128) are depicted as circles. **e**, Crystal structure of Ubiquitin-E2 conjugate (PDB ID 7r71) in a closed conformation superimposed onto the cryo-EM structure of UREL:UFM1-bound 60S ribosome in the same view as (**d**). C-terminal glycine of ubiquitin (G76), catalytic cysteine to lysine mutation C85K of ubiquitin E2 (UBE2D2) and most C-terminal residue of RPL26 for which density is present in the cryo-EM map (V128) are depicted as circles. **f**, Quantification of immunoblots from (Figure 5b). UFMylated RPL26 band intensity normalized to intensity of total RPL26 (left) and SEC61β band intensities normalized to RPL10a. Data show mean ± SD. ***$p < 0.0001$ (Student t-test). Data are representative of $n = 2$ independent experiments. **g**, UFMylation mediates dissociation of 60S from the translocon. *In vitro* UFMylation reactions were performed on membrane associated 60S ribosomal subunit-SEC61 translocon complexes isolated from CDK5RAP3 KO cells. 60S-SEC61 complexes were incubated with UBA5, UFC1, UREL and UFM1 either in the presence or absence of ATP and the reaction products were separated on a sucrose gradient and analysed by immunoblotting with the indicated antibodies. Blot is representative of $n = 2$ independent experiments. **h**, Dissociation of 60S from the ER translocon requires the UFIM motif. *In vitro* 60S-SEC61 UFMylation and translocon dissociation assay performed as in (**d**) using UREL complex with the UFBP1 UFIM mutant F196A. Blot is representative of $n = 2$ independent experiments. **i**, Comparison of the UFL1 PTC loop (P-site) with the eRF1 catalytic centre (A-site; PDB ID 6ip8). eRF1 catalytic residues GGQ and UFL1 loop residues GGN coloured in yellow. **j**, Superimposition of UREL complex and NEMF:Listerin complex (PDB ID 3j92) bound to the 60S ribosome. Missing Listerin model is depicted as dashed line.

Extended Data Table 1 | Summary of Cryo-EM data collection, processing and model statistics

| Data collection parameters | Ligase bound 60S ribosome | Ligase complex only | UFMylated 60S ribosome |
|---|---|---|---|
| Microscope | FEI Titan Krios | | |
| Detector | Thermo Scientific Falcon4i | | Thermo Scientific Falcon4 |
| Energy filter | Thermo Scientific SelectrisX | | - |
| Energy filter slit (eV) | 10 | | - |
| Accelerating Voltage (kV) | 300 | | |
| Magnification | x165,000 | | x96,000 |
| Spot size | 8 | | |
| Illuminated area (mm) | 0.54 | | 0.56 |
| Pixel size (Å) | 0.74 | | 0.82 |
| Defocus range (mm) | -0.2 to -2.0 | | -0.8 to -2.9 |
| Total electron dose (e⁻/Å²) | 33.4 | | 39.8 |
| Exposure (s) | 2.67 | | 6.66 |
| Number of frames | 42 | | 32 |
| Dose per frame (e⁻/Å²) | 0.8 | | 0.8 |
| Number of micrographs | 59,394 | | 3,028 |
| Acquisition mode | Counting | | |
| AFIS | Yes | | No |
| AFIS range (mm) | 8 | | - |
| EPU software version | 3.2.0 | | 2.14 |
| **Data possessing parameters** | **Ligase bound 60S ribosome** | **Ligase complex only** | **UFMylated 60S ribosome** |
| Symmetry point group | C1 | | |
| Final particle number | 299,008 | | 5,402 |
| Map resolution (Å) | 2.2 | 3.2 | 7.0 |
| FSC threshold | 0.143 | | |
| Map sharpening B factor (Å²) | 59.28 | - | - |
| **Model statistics** | **Ligase bound 60S ribosome** | **Ligase complex only** | |
| Bond length (Å) (# > 4σ) | 0.004 | 0.004 | |
| Bond angles (°) (# > 4σ) | 0.595 | 0.565 | |
| MolProbity score | 1.65 | 1.82 | |
| Clash score | 2.04 | 9.25 | |
| Ramachandran outliers (%) | 0.02 | 0.00 | |
| Ramachandran allowed (%) | 1.56 | 4.77 | |
| Ramachandran favoured (%) | 98.43 | 95.23 | |
| Rotamer outliers (%) | 0.95 | 0.09 | |
| Cβ outliers (%) | 0.00 | 0.00 | |
| Cis proline (%) | 4.5 | 0.0 | |
| CaBLAM outliers (%) | 1.12 | 2.63 | |

**Extended Data Table 2 | Summary of X-ray crystallography data collection and refinement statistics**

| | UFC1-UFM1 (PDB 8bzr) | UFL1/UFBP1/UFC1 (PDB 8c0d) |
|---|---|---|
| **Data collection** | | |
| Space group | $P22_12_1$ | $P2_12_12_1$ |
| $a, b, c$ (Å) | 56.00, 56.43,86.96 | 79.21, 124.57, 131.29 |
| $\alpha, \beta, \gamma$ (°) | 90.00, 90.00, 90.00 | 90.00, 90.00, 90.00 |
| Resolution range (Å) | 47.33-1.78 (1.81-1.78) | 65.65-2.56 (2.89-2.56) |
| Total No. of reflections | 335,091 (8,900) | 330,110 (17,196) |
| No. of unique reflections | 26,839 (1,192) | 25,858 (1,293) |
| Completeness (%) | 99.1 (88.6) | 93.8 (74.1) |
| Multiplicity | 12.5 (7.5) | 12.8 (13.3) |
| $\langle I/\sigma(I) \rangle$ | 9.2 (0.5) | 8.2 (1.6) |
| $CC_{1/2}$ | 0.998 (0.270) | 0.997 (0.674) |
| **Refinement** | | |
| No. of reflections, working set | 26,800 | 24,636 |
| No. of reflections, test set | 1,360 | 1,222 |
| Final $R_{work}$ | 0.182 | 0.232 |
| Final $R_{free}$ | 0.226 | 0.286 |
| No. of non-H atoms | | |
| Protein | 1,943 | 12,700 |
| Ligand | 12 | 0 |
| Water | 106 | 7 |
| Total | 2,061 | 12,707 |
| R.m.s. deviations | | |
| Bonds (Å) | 0.0106 | 0.0105 |
| Angles (°) | 1.726 | 1.888 |
| Average $B$-factors (Å$^2$) | 35.0 | 51.0 |

# Reporting Summary

## Statistics

For all statistical analyses, confirm that the following items are present in the figure legend, table legend, main text, or Methods section.

| n/a | Confirmed | |
|---|---|---|
| ☐ | ☒ | The exact sample size (*n*) for each experimental group/condition, given as a discrete number and unit of measurement |
| ☐ | ☒ | A statement on whether measurements were taken from distinct samples or whether the same sample was measured repeatedly |
| ☐ | ☒ | The statistical test(s) used AND whether they are one- or two-sided<br>*Only common tests should be described solely by name; describe more complex techniques in the Methods section.* |
| ☒ | ☐ | A description of all covariates tested |
| ☒ | ☐ | A description of any assumptions or corrections, such as tests of normality and adjustment for multiple comparisons |
| ☐ | ☒ | A full description of the statistical parameters including central tendency (e.g. means) or other basic estimates (e.g. regression coefficient) AND variation (e.g. standard deviation) or associated estimates of uncertainty (e.g. confidence intervals) |
| ☐ | ☒ | For null hypothesis testing, the test statistic (e.g. *F*, *t*, *r*) with confidence intervals, effect sizes, degrees of freedom and *P* value noted<br>*Give P values as exact values whenever suitable.* |
| ☒ | ☐ | For Bayesian analysis, information on the choice of priors and Markov chain Monte Carlo settings |
| ☒ | ☐ | For hierarchical and complex designs, identification of the appropriate level for tests and full reporting of outcomes |
| ☒ | ☐ | Estimates of effect sizes (e.g. Cohen's *d*, Pearson's *r*), indicating how they were calculated |

*Our web collection on statistics for biologists contains articles on many of the points above.*

## Software and code

Policy information about availability of computer code

| Data collection | Yes described and referenced in Methods section. |
|---|---|
| Data analysis | Yes, described and referenced in Methods section. Crystallography data was processed with the autoPROC suite (including XDS62, Pointless, Aimless, CCP4 and STARANISO or Xia2 and DIALS. Molecular replacement - PHASER. Refinement and model building- REFMAC and Coot (CCP4i2 suite). Cryo-EM data processing - MOTIONCOR2 and CTFFIND4.1 (RELION-3.1), crYOLO-1.6.1, cryoSPARC-3.2, cryoSPARC-4.2.1, cryoDRGN-3.2.0, cryoSPARC-4.2.1 3DFlex. Model building, map sharpening and refinement - Phenix-1.2.1, DeepEMhancer, Coot-0.9.8.1. Visualization - UCSF ChimeraX-1.2.5 software |

For manuscripts utilizing custom algorithms or software that are central to the research but not yet described in published literature, software must be made available to editors and reviewers. We strongly encourage code deposition in a community repository (e.g. GitHub). See the Nature Portfolio guidelines for submitting code & software for further information.

## Data

Policy information about availability of data

All manuscripts must include a data availability statement. This statement should provide the following information, where applicable:

- Accession codes, unique identifiers, or web links for publicly available datasets
- A description of any restrictions on data availability
- For clinical datasets or third party data, please ensure that the statement adheres to our policy

The cryo-EM coordinates have been deposited to the Protein Data Bank under accession codes 8QFD (ligase bound 60S) and 8QFC (ligase only). Cryo-EM maps have been deposited to the Electron Microscopy Data Bank under accession codes EMD-18382 (ligase bound 60S) and EMD-18381 (ligase only).
X-ray structure factors and associated models have been deposited to the Protein Data Bank under accession codes 8C0D and 8BZR. The Raw DSBU XL-MS data and the LC-MS/MS analysis of ribosome UFMylation have been deposited to the ProteomeXchange Consortium via the PRIDE partner repository with the dataset identifiers PXD046990 and PXD046991, respectively.

## Research involving human participants, their data, or biological material

Policy information about studies with human participants or human data. See also policy information about sex, gender (identity/presentation), and sexual orientation and race, ethnicity and racism.

| | |
|---|---|
| Reporting on sex and gender | N/A |
| Reporting on race, ethnicity, or other socially relevant groupings | N/A |
| Population characteristics | N/A |
| Recruitment | N/A |
| Ethics oversight | N/A |

Note that full information on the approval of the study protocol must also be provided in the manuscript.

# Field-specific reporting

Please select the one below that is the best fit for your research. If you are not sure, read the appropriate sections before making your selection.

☒ Life sciences ☐ Behavioural & social sciences ☐ Ecological, evolutionary & environmental sciences

For a reference copy of the document with all sections, see nature.com/documents/nr-reporting-summary-flat.pdf

# Life sciences study design

All studies must disclose on these points even when the disclosure is negative.

| | |
|---|---|
| Sample size | Where necessary, we have repeated experiments and the number of biological and technical replicates are indicated in the figure legends. In our experience for cell-based assays in cell lines (n>3) and in vitro assays using purified proteins (n>2), reproducibility with these numbers are sufficient. |
| Data exclusions | No data was excluded |
| Replication | In general, we aimed for sufficient technical and biological replicates as indicated in each figure legend to ensure robust reproducibility of observations |
| Randomization | No treatments were performed on different biological samples where human bias could be a factor, randomization was not applicable. |
| Blinding | No treatments were performed on different biological samples where human bias could be a factor, blinding was not applicable |

# Reporting for specific materials, systems and methods

We require information from authors about some types of materials, experimental systems and methods used in many studies. Here, indicate whether each material, system or method listed is relevant to your study. If you are not sure if a list item applies to your research, read the appropriate section before selecting a response.

## Materials & experimental systems

| n/a | Involved in the study |
|-----|----------------------|
| ☐ | ☒ Antibodies |
| ☐ | ☒ Eukaryotic cell lines |
| ☒ | ☐ Palaeontology and archaeology |
| ☒ | ☐ Animals and other organisms |
| ☒ | ☐ Clinical data |
| ☒ | ☐ Dual use research of concern |
| ☒ | ☐ Plants |

## Methods

| n/a | Involved in the study |
|-----|----------------------|
| ☒ | ☐ ChIP-seq |
| ☒ | ☐ Flow cytometry |
| ☒ | ☐ MRI-based neuroimaging |

# Antibodies

| | |
|---|---|
| Antibodies used | Anti-UFM1 Abcam ab109305<br>Anti-UBA5 Universal Biologicals A304-115A-T<br>Anti-UFC1 Abcam ab189252<br>Anti-UFL1 Abcam ab227506<br>Anti-UFBP1 Abcam ab99121<br>Anti-CDK5RAP3 Bethyl Laboratories A300-870A<br>Anti-RPL26 Bethyl laboratories A300-686A-M<br>Anti-rabbit IgG, HRP-linked Antibody CST 70745<br>IRDye 800CW anti-Rabbit Li-COR 926-32211<br>IRDye 680CW anti-Rabbit Li-COR 926-68071<br>Anti-UFBP1 (DDRGK1) Proteintech 21445-1-AP<br>Anti-CDK5RAP3 Proteintech 11007-1-AP<br>Anti-UFSP2 Abcam ab192597<br>Anti-RPL26 Abcam ab59567<br>Anti-ERp72 CST 5033T<br>Anti-GAPDH Abcam ab8245<br>Anti-RPS10 Abcam ab151550<br>Anti-RPL10A Abcam ab174318<br>Anti-SEC61-b (gift from R.Hegde, MRC LMB Cambridge) |
| Validation | Since most of the antibodies listed here were used in biochemical assays, the specificity of the antibody was established by having conditions where the protein was not present and a signal was not detected at the corresponding molecular weight by immunoblotting. Specificity statements are also available on the manufacturer's website:<br>Anti-UFM1 – See specificity statement from the manufacturer's website - https://www.abcam.com/products/primary-antibodies/ufm1-antibody-epr42642-ab109305.html<br><br>Anti-UBA5 - See specificity statement from the manufacturer's website - https://www.thermofisher.com/antibody/product/UBA5-Antibody-Polyclonal/A304-115A-T<br><br>Anti-UFC1 - See specificity statement from manufacturer's website. - https://www.abcam.com/products/primary-antibodies/ufc1-antibody-epr15014-102-ab189252.html<br><br>Anti-UFL1 - See specificity statement from manufacturer's website – https://www.abcam.com/products/primary-antibodies/ufl1-antibody-n-terminal-ab227506.html<br><br>Anti-CDK5RAP3 – See specificity statement from manufacturer's website – https://www.thermofisher.com/antibody/product/CDK5RAP3-Antibody-Polyclonal/A300-870A<br><br>Anti-DDRGK1 - See specificity statement from manufacturer's website - https://www.ptglab.com/products/DDRGK1-Antibody-21445-1-AP.htm<br><br>Anti-UFSP2 - See specificity statement from manufacturer's website - https://www.abcam.com/products/primary-antibodies/ufsp2-antibody-ep13424-49-ab192597.html<br><br>Anti-RPL26 - See specificity statement from manufacturer's website - https://www.abcam.com/products/primary-antibodies/rpl26-antibody-ab59567.html<br>For the anti-Sec61b antibody, it was validated in the publication from the Hegde lab. for example - doi:10.1083/jcb.200210095 |

# Eukaryotic cell lines

Policy information about cell lines and Sex and Gender in Research

| | |
|---|---|
| Cell line source(s) | Flp-In T-REx HEK293 cells (Invitrogen; R78007) |
| Authentication | None |

| Mycoplasma contamination | Cell lines were routinely tested for mycoplasma and were negative for mycoplasma contamination |
| Commonly misidentified lines (See ICLAC register) | Not used |

