## [Peer Review File · Nature]

Manuscript Title: The UFM1 E3 ligase recognizes and releases 60S ribosomes from ER translocons

Redactions – unpublished data

Reviewer Comments & Author Rebuttals

Reviewer Reports on the Initial Version:

Referee #1: cryo-EM, ribosome

Referee #2: cryo-EM, protein quality control, translation

Referees' comments:

Referee #1 (Remarks to the Author):

A. Summary of Key Results

Makhlouf et al. provide structural insights into the recognition of the 60S ribosomal substrate and its ufmylation by the UFM1-E3-Ligase complex. The authors present a novel structure of the Ligase deeply inserted into the peptidase transfer center of the ribosomal substrate. While these insights are compelling, the manuscript has several technical issues that need to be addressed.

B. Originality and Significance

The study is indeed original, providing intriguing insights into the recognition of the ribosomal substrate by the Ligase. The deep insertion of a loop into the peptidase center is particularly remarkable. The exploration of the UFM1 transfer mechanism is also of note, particularly since the ligase does not resemble classical ubiquitin E3 ligases. The study holds potential interest for multiple communities including those studying ribosomes, ubiquitin, and protein quality control.

C. Data and Methodology

The cryoEM structure of the ligase bound to the 60S particle is the crux of this study. However, the quality of this structure is not up to the standard required for this study. The structure exhibits signs of overfitting, as evidenced by the spikey appearance of the E3 portion density, leading to incorrect side chain fittings and potentially faulty conclusions. Also the occupancy of the ligase overall is low maybe due to flexibility. Thus, the overall density is fragmented, with significant portions of the ligase difficult to interpret.

While the authors' honesty in acknowledging these issues is appreciated, there is a clear need for a more complex analysis scheme. The authors use a single classification step and one focus step. In this focus step the mask was not cleanly made as it included blobs of ribosomal density detached from the ligase itself. I suggest improving the cryoEM density with a modern sorting scheme and maybe even including sophisticated tools such as cryodrgn, 3D Flex, and various focused or multibody refinements. I took the liberty to run DeepEMhancer on the provided maps and could

already see that there is a huge improvement thus I believe more dedicated work on the density will yield good results.

Despite the low map quality, the model building is largely meticulous, but it still misses out on modeling some regions with good density, such as the RNA of the L1 stalk, which should be modelled as well.

Over-interpretation should be avoided where density does not provide sufficient evidence, and invisible parts such as side chains could be removed or set to occupancy values of 0.

The study also includes several mutational studies and quantitative binding assays, which are well-executed but could be further expanded upon.

D. Appropriate Use of Statistics and Treatment of Uncertainties

The report lacks sufficient information about how often experiments were repeated and whether they were repeated at all. Given the scant details provided in the reporting summary, it's essential that this issue is addressed and all experiments are confirmed to have been repeated at least once.

E. Conclusions: Robustness, Validity, Reliability

Given the flawed cryoEM density of the UREL and the potential errors in the model, the conclusions of the study may not be robust and valid. More claims should be supported by mutagenesis, especially those concerning the intriguing loop in the PTC. The proposed E2 shuttling mechanism and the claim that the E3 itself is a reader are somewhat speculative and require further investigation.

F. Suggested Improvements: Experiments, Data for Possible Revision

1. Improve the structure as mentioned above.
2. Investigate the significance and specificity of the loop insertion with deletions or isolated peptides.
3. Conduct further investigations on UFMylation of the 80S, which seems to be a loose end that is not addressed in later parts of the study anymore.
4. Address the role of the membrane context in the process. The reactions clearly happen in the context of the ER membrane while all experiments are done with free 60S. It should be at least discussed clearly what this would mean for the mechanism and the validity of the experiments done.

G. References: Appropriate Credit to Previous Work?

The early works on UFM1, including its discovery in the Tanaka lab, are missing and should certainly be included.

H. Clarity and Context: Lucidity of Abstract/Summary, Appropriateness of Abstract, Introduction, and

Conclusions

While the abstract and summary are appropriate, but the manuscript is quite dense and could benefit from clearer and more streamlined writing. Also the overall model figure should go to the main figure and not be hidden in the extended data. More of such cartoon figures in general will aid the clarity of the manuscript.

Overall I think this manuscript should get the chance to go through a revision.

Referee #2 (Remarks to the Author):

The study by Makhlof, Peter, Magnussen reports a cryo-EM structure of the UFM1 E3 ligase complex (UREL) bound to 60S ribosomal subunits and a crystal structure of a minimal E3-UFC1 (the cognate E2 enzyme) complex. Considered together with mutagenesis, interaction studies, and functional assays, the authors propose a model for UFMylation on RPL26, a specific modification on ribosomes with poorly understood but physiologically important consequences. Overall, the structural and biochemical work are of high quality and reveal interesting observations about how UREL interacts with the 60S ribosomal subunit. This includes how the complex precludes translation, has several intriguing similarities with other ribosome-binding factors, and appears to be both a 'writer' and 'reader' of UFMylation. Obtaining a structure of this complex is a notable achievement. However, despite attempts to gain clearer understanding into the mechanism of UFMylation, the insights from this study remain fairly descriptive and some interpretations are incompletely substantiated.

Specific comments:

1. In Extended Data Fig. 3k, the sequence similarity does not seem to match structural overlap. Where are the conserved residues of UFBP1 and CCDC47 located in the structures?
2. The interpretations related to UFBP1 and CCDC47 also seem counterintuitive since PAT complex engagement does not seem to disrupt Sec61 binding (PMID: 36261528), unlike the proposal here that UREL binding disrupts Sec61 interactions. Since Sec61 does not appear to be present in the samples analyzed structurally here, is there functional data supporting this claim?
3. The comparison of the GGQ motif of eRF1 and the GGN motif of the UFL1 PTC loop is interesting, but the locations of the two motifs are not so similar. It is not clear what insights one can take away from this beyond the interesting observation.
4. The sentence "Intriguingly the remodelling of the PTC bears striking resemblance to the translation of arrest peptides or antibiotics such as anisomycin, which have also been known to remodel key PTC bases in order to induce ribosome stalling" seems overstated. Since there is no ongoing translation, these interactions cannot stall translation. From Extended Data Fig. 4, it is also not clear what the "striking resemblance(s)" are besides the observation that these elements all bind in a similar place near the PTC – are the remodeling of the bases at the PTC as described in lines 209-215 conserved with these other compounds? If so, please show this.
5. I found the XL-MS (Extended Data Fig. 6) and biochemical data (Fig. 5e, f) supporting a second

UFC1-UFM1 binding site confusing to interpret. The XL-MS plots do not appear to provide sufficient information/resolution to know how justified the presented structural models are. It would be useful to see the contact sites highlighted in structural models showing both UFC1-UFM1 binding sites on the E3. Can the CDK5RAP3-dependent interaction between UFC1-UFM1 and E3mUU be recapitulated with pulldowns or another assay that permits more direct comparisons between different conditions? The separation of UFL1/UFBP1 and CDK5RAP3 in the SEC traces is not so clear, with some spurious peaks/protein migration patterns that appear to be difficult to explain. How reproducible are these migration patterns?

Minor comments:

1. Extended Fig. 3j: what happens to the levels of other UREL and UFMylation factors after knocking out CDK5RAP3?
2. Not all figure panels are called out and some are occasionally called out of order or may be referred to incorrectly (particularly the discussion around Fig. 5e,f and Extended Data Fig. 7-9). Some reordering/arranging would help readability.

Author Rebuttals to Initial Comments:

We thank the reviewers for their interest in our work and for their constructive comments and suggestions, which have strengthened our findings and improved the manuscript. We have provided a point-by-point response addressing each comment.

Referee #1 (Remarks to the Author):

A. Summary of Key Results

Makhlouf et al. provide structural insights into the recognition of the 60S ribosomal substrate and its ufmylation by the UFM1-E3-Ligase complex. The authors present a novel structure of the Ligase deeply inserted into the peptidase transfer center of the ribosomal substrate. While these insights are compelling, the manuscript has several technical issues that need to be addressed.

B. Originality and Significance

The study is indeed original, providing intriguing insights into the recognition of the ribosomal substrate by the Ligase. The deep insertion of a loop into the peptidase center is particularly remarkable. The exploration of the UFM1 transfer mechanism is also of note, particularly since the ligase does not resemble classical ubiquitin E3 ligases. The study holds potential interest for multiple communities including those studying ribosomes, ubiquitin, and protein quality control.

We thank the reviewer for recognising the importance and originality of our findings and the potential for our manuscript to appeal to a broad scientific audience.

C. Data and Methodology

The cryoEM structure of the ligase bound to the 60S particle is the crux of this study. However, the quality of this structure is not up to the standard required for this study. The structure exhibits signs of overfitting, as evidenced by the spikey appearance of the E3 portion density, leading to incorrect side chain fittings and potentially faulty conclusions.

Also the occupancy of the ligase overall is low maybe due to flexibility. Thus, the overall density is fragmented, with significant portions of the ligase difficult to interpret. While the authors' honesty in acknowledging these issues is appreciated, there is a clear need for a more complex analysis scheme. The authors use a single classification step and one focus step. In this focus step the mask was not cleanly made as it included blobs of ribosomal density detached from the ligase itself. I suggest improving the cryoEM density with a modern sorting scheme and maybe even including sophisticated tools such as cryodrgn, 3D Flex, and various focused or multibody refinements. I took the liberty to run DeepEMhancer on the provided maps and could already see that there is a huge improvement thus I believe more dedicated work on the density will yield good results.

Despite the low map quality, the model building is largely meticulous, but it still misses out on modeling some regions with good density, such as the RNA of the L1 stalk, which should be modelled as well.

We have now improved our model and included the RNA in L1 stalk. This was greatly aided by the improved maps obtained using the suggestions, and we thank the reviewer for this guidance.

After focussed refinement of the ligase region, we used 3DFlex which revealed significant flexibility in the catalytic region of the ligase. The 3DFlex reconstruction markedly improved the local resolution of this area from ~6 Å to 4 Å.

Figure: Cryo-EM map of the UREL ligase complex after focussed refinement (a) and after 3DFlex reconstruction (b), coloured by local resolution. FSC curve is shown, calculated using the gold-standard FSC cut-off at 0.143.

The 3D flex analyses have also revealed multidirectional motions of the catalytic module, which we believe provides additional valuable information about the UFMylation mechanism and how the ligase works, and also explains why the local resolution is low in this area. We are grateful to the reviewer for this suggestion and have added these new analyses in **Fig. 4 b** and **Supplementary Movie 2**.

We also thank the reviewer for their suggestion to use DeepEMhancer which greatly improved the interpretability of the maps. However, despite noticeable improvements in the quality of the maps, including the L1 stalk and many important ligase-60S interacting regions, we were still unable to build side chains in some low-resolution regions and have acknowledged this clearly in the manuscript. We also attempted further classification steps such as cryoDRGN

which removed some junk particles but unfortunately, this did not improve the density for the ligase.

Over-interpretation should be avoided where density does not provide sufficient evidence, and invisible parts such as side chains could be removed or set to occupancy values of 0.

We have set occupancy values to 0 in regions where density is not sufficient for side chain modelling. We have also indicated the model building limitation in the methods and figure legends.

The study also includes several mutational studies and quantitative binding assays, which are well-executed but could be further expanded upon.

Please see below for experiments we have performed to gain more insights using different mutants.

D. Appropriate Use of Statistics and Treatment of Uncertainties

The report lacks sufficient information about how often experiments were repeated and whether they were repeated at all. Given the scant details provided in the reporting summary, it's essential that this issue is addressed and all experiments are confirmed to have been repeated at least once.

We have now included this information for all the data presented in the manuscript.

E. Conclusions: Robustness, Validity, Reliability

Given the flawed cryoEM density of the UREL and the potential errors in the model, the conclusions of the study may not be robust and valid. More claims should be supported by mutagenesis, especially those concerning the intriguing loop in the PTC. The proposed E2 shuttling mechanism and the claim that the E3 itself is a reader are somewhat speculative and require further investigation.

We thank the reviewer for their suggestions in section F which are related to this section, and we have addressed in more detail (please see below).

F. Suggested Improvements: Experiments, Data for Possible Revision

1. Improve the structure as mentioned above.

We thank the reviewer for their constructive comments. We greatly appreciate that they have looked at our maps and model carefully and are grateful for indicating that DeepEMhancer can improve our maps. We agree with the reviewer that the new processing tools can lead to improvements, and we have now used cryoDRGN, 3DFlex and DeepEMhancer as suggested.

This significantly improved the overall structure and has led to new insights about the flexibility of the ligase complex.

2. Investigate the significance and specificity of the loop insertion with deletions or isolated peptides.

We obtained biotinylated peptides corresponding to the loop insertion and performed pull down experiments to check for binding to 60S. However, a control peptide with single point mutations of interacting residues in the loop also pulls down ribosomes. We remain cautious of this approach as the very positively charged peptide may lead to non-specific interactions.

Redacted

At present, we do not understand the function of this loop insertion and are planning to investigate this in detail in future research. This will revolve around our hypothesis that the Asn in the GGN motif of the PTC loop may work similarly to the Gln in the GGQ motif of eRF1 to mediate peptide-tRNA hydrolysis. However, we feel this is beyond the scope of the present manuscript.

3. Conduct further investigations on UFMylation of the 80S, which seems to be a loose end that is not addressed in later parts of the study anymore.

4. Address the role of the membrane context in the process. The reactions clearly happen in the context of the ER membrane while all experiments are done with free 60S. It should be at least discussed clearly what this would mean for the mechanism and the validity of the experiments done.

We agree this is an important point. We performed in vitro UFMylation assays in membrane fractions and found that 60S is efficiently UFMylated. Our cryoEM structure showed that UFBP1's N-terminal helical arm is bound to the ribosome in a way that it may promote dissociation of SEC61 translocon. This prompted us to test the hypothesis that a primary function of UFMylation is to dissociate 60S subunits from SEC61. If this were true, then in cells lacking functional UREL, there would be an accumulation of 60S-SEC61 complexes. Indeed, we observe striking co-sedimentation of SEC61 with 60S in CDK5RAP3 knockout HEK293 cells and not in WT cells. We then used the 60S-SEC61 complexes isolated from the membrane fractions of CDK5RAP3 knockout cells to test if UFMylation would lead to disassociation of SEC61 from 60S subunits. Excitingly, we find that disassociation only occurs in the presence of functional UFMylation and additionally requires the "reader" function of UREL.

We thank the reviewer for the suggestion as it prompted us to do these experiments that provided these important new insights.

G. References: Appropriate Credit to Previous Work?

The early works on UFM1, including its discovery in the Tanaka lab, are missing and should certainly be included.

We apologize for the oversight, and we have now included references (Ref. Nr 6 and 7 of revised manuscript) to the original discovery of UFM1 and UFL1.

H. Clarity and Context: Lucidity of Abstract/Summary, Appropriateness of Abstract, Introduction, and Conclusions

While the abstract and summary are appropriate, but the manuscript is quite dense and could benefit from clearer and more streamlined writing. Also the overall model figure should go to the main figure and not be hidden in the extended data. More of such cartoon figures in general will aid the clarity of the manuscript.

We have now extensively edited the manuscript text and reorganized the figures, which we think has improved the manuscript. We have also moved the overall model figure to the main figure and added cartoons to improve clarity.

Overall I think this manuscript should get the chance to go through a revision.
We thank the reviewer

Referee #2 (Remarks to the Author):

The study by Makhlof, Peter, Magnussen reports a cryo-EM structure of the UFM1 E3 ligase complex (UREL) bound to 60S ribosomal subunits and a crystal structure of a minimal E3-UFC1 (the cognate E2 enzyme) complex. Considered together with mutagenesis, interaction studies, and functional assays, the authors propose a model for UFMylation on RPL26, a specific modification on ribosomes with poorly understood but physiologically important consequences. Overall, the structural and biochemical work are of high quality and reveal interesting observations about how UREL interacts with the 60S ribosomal subunit. This includes how the complex precludes translation, has several intriguing similarities with other ribosome-binding factors, and appears to be both a 'writer' and 'reader' of UFMylation. Obtaining a structure of this complex is a notable achievement. However, despite attempts to gain clearer understanding into the mechanism of UFMylation, the insights from this study remain fairly descriptive and some interpretations are incompletely substantiated.

We thank the reviewer for commending the high quality of our work and for recognising the importance of the structure of this complex.

Specific comments:

1. In Extended Data Fig. 3k, the sequence similarity does not seem to match structural overlap. Where are the conserved residues of UFBP1 and CCDC47 located in the structures?
2. The interpretations related to UFBP1 and CCDC47 also seem counterintuitive since PAT complex engagement does not seem to disrupt Sec61 binding (PMID: 36261528), unlike the proposal here that UREL binding disrupts Sec61 interactions. Since Sec61 does not appear to be present in the samples analyzed structurally here, is there functional data supporting this claim?

We agree with the reviewer that the sequence similarity does not match the structural overlap. The point we were trying to make is that despite UFBP1 and CCDC47 binding to similar regions, only UREL binding disrupts interactions with SEC61. We have now reworded this section to remove this which has improved clarity and avoided speculation.

While SEC61 is not present in the samples analyzed structurally here, this suggestion from the reviewer prompted us to examine the effect of UREL and UFMylation on 60S-SEC61 association. Indeed, we find that SEC61 sediments together with the 60S fraction in CDK5RAP3 knockout cells. We then set up an in vitro reconstitution system that demonstrated

a fundamental function for UFMylation in splitting SEC61 from 60S subunits. We have included these results in **Fig. 6** of the revised manuscript.

3. The comparison of the GGQ motif of eRF1 and the GGN motif of the UFL1 PTC loop is interesting, but the locations of the two motifs are not so similar. It is not clear what insights one can take away from this beyond the interesting observation.

The location of the two motifs correlates with the A-site for eRF1 and P-site for UFL1 PTC loop which explains why the locations of the two motifs are not so similar. For now, this is an interesting observation and we do not know the functional consequences of this loop insertion or the GGN motif. Given the similarity, we speculate that the Asn in the GGN motif works similarly to the Gln in the GGQ motif of eRF1 to mediate peptide-tRNA hydrolysis. Alternatively, it could be a mechanism of molecular mimicry to survey vacant PTC. This is something we plan to investigate in detail in the future.

4. The sentence “Intriguingly the remodelling of the PTC bears striking resemblance to the translation of arrest peptides or antibiotics such as anisomycin, which have also been known to remodel key PTC bases in order to induce ribosome stalling” seems overstated. Since there is no ongoing translation, these interactions cannot stall translation. From Extended Data Fig. 4, it is also not clear what the “striking resemblance(s)” are besides the observation that these elements all bind in a similar place near the PTC – are the remodeling of the bases at the PTC as described in lines 209-215 conserved with these other compounds? If so, please show this.

We agree with the reviewer that the PTC loop is not likely to induce ribosome stalling since there is no ongoing translation. Remodelling of the bases at the PTC is not conserved and the arrest peptides and antibiotics shown in Extended Data Fig. 4 stall the ribosome via different mechanisms. We simply wanted to highlight the interesting observation that the PTC loop and stalling agents bind to or near the PTC and influence the position of the surrounding bases.

We have rewritten this section (lines 228-232) and removed the original text that could be misconstrued to suggest a role for the PTC loop in inducing ribosome stalling. “Antibiotics such as anisomycin and translated arrest peptides also bind to this region and remodel the PTC bases (Extended Data Fig. 4d). The remodelled state of the PTC we observe bears resemblance to the PTC conformations observed in structures of ribosomes translating arrest peptides or in complex with anisomycin²².”

5. I found the XL-MS (Extended Data Fig. 6) and biochemical data (Fig. 5e, f) supporting a second UFC1-UFM1 binding site confusing to interpret. The XL-MS plots do not appear to provide sufficient information/resolution to know how justified the presented structural models are. It would be useful to see the contact sites highlighted in structural models showing both UFC1-UFM1 binding sites on the E3.

We have now shown the contact sites of UFC1~UFM1 on the structural models of the E3 (**Extended Data Fig. 8**), which we hope makes it clearer to see the presence of the second UFC1-UFM1 site. We have also performed an additional experiment (**Fig. 5f**) that we believe demonstrates this (see below).

Can the CDK5RAP3-dependent interaction between UFC1-UFM1 and E3mUU be recapitulated with pulldowns or another assay that permits more direct comparisons between different conditions? The separation of UFL1/UFBP1 and CDK5RAP3 in the SEC traces is not so clear, with some spurious peaks/protein migration patterns that appear to be difficult to explain. How reproducible are these migration patterns?

These results are very reproducible, observed in over 5 experiments and so we are confident of this CDK5RAP3-dependent interaction. We have now represented the SEC traces in such a way that it is clearer and easier to tell the differences. We have also run the individual components separately for further clarity (**Extended Data Fig. 9**). Despite these efforts, we still see a double peak where the first peak corresponds to the complex with CDK5RAP3 and the second peak is the E3mUU:UFC1-UFM1 complex without CDK5RAP3. This might make it difficult for the reader to interpret the CDK5RAP3-dependent interaction between UFC1-UFM1 and E3mUU.

Hence, we have also designed a new experiment to prove this CDK5RAP3-dependent interaction. We used full length UFL1, UFBP1 and took advantage of the fact that UFC1 interacts with the N-terminal helix of UFL1. Hence, UFL1 Δ helix/UFBP1 can no longer bind to UFC1~UFM1. Now, when CDK5RAP3 is added, a complex containing UFL1 Δ helix/UFBP1, CDK5RAP3 and UFC1~UFM1 is formed, providing further evidence of this additional UFC1~UFM1 binding site on UREL. This result is shown in **Fig. 5g** of the revised manuscript.

Minor comments:

1. Extended Fig. 3j: what happens to the levels of other UREL and UFMylation factors after knocking out CDK5RAP3?

We have immunoblotted for the expression levels of UFL1, UFBP1, CDK5RAP3, UFM1, UFSP2 and ODR4 in WT and CDK5RAP3 knock out cells and find that the levels of the UFMylation components are unchanged. These results have now been included in **Extended Data Fig. 3j**.

2. Not all figure panels are called out and some are occasionally called out of order or may be referred to incorrectly (particularly the discussion around Fig. 5e,f and Extended Data Fig. 7-9). Some reordering/arranging would help readability.

We thank the reviewer for pointing this out and we have reorganized the figures to fix this and have also rearranged figures to fit the flow of the narrative.

Reviewer Reports on the First Revision:

Referees' comments:

Referee #1 (Remarks to the Author):

The authors have diligently addressed my previous comments, enhancing the clarity and robustness of the manuscript. I appreciate their efforts in refining the structure and ensuring the integrity of their work. I do have a few small additional suggestions/ requests I would ask the authors to implement.

1. In the ligase-only structure, there is a minor density blob outside the ligase (see attached image) that seems to originate from the mask, and doesn't contain meaningful density. I suggest that removing this might enhance the quality of the map. Please rerefine without this meaningless blob. I am sure it will improve the overall map quality further.

2. The clashes depicted in Figure 1f aren't immediately evident from the visual representation provided. The UFBP1 helix appears to fit well into the sec61 channel in the given image. I recommend using surface rendering for both components to clarify the point of contention. Additionally, quantifying the overlap could offer a clearer perspective.

3. The inclusion of angular distribution plots, which are standard in the cryo-EM field, would be beneficial.

4. In line with best scientific practices, I suggest the authors make their raw data available on EMPIAR.

Outside of this, I highly recommend this manuscript to be published in Nature!

Referee #2 (Remarks to the Author):

The revised manuscript addresses most of my original concerns. My remaining comment relates, still, to the interpretation of the GGN motif, which I continue to find to be somewhat imprecise and overly speculative in the absence of do-able experiments. Specifically:

A. To my knowledge, the exact mechanism of how the GGQ motif of release factors catalyzes peptidyl-tRNA hydrolysis is not fully understood at a molecular level. It is thought that this motif works with rRNA bases of the peptidyl-transferase center (PTC) to coordinate a water molecule that can act as a nucleophile to hydrolyze the peptidyl-tRNA ester bond. I agree it is tempting to speculate that what the authors modeled may represent some sort of intermediate (e.g. post-hydrolysis) of this process. However, considering how far away the GGN sequence in their structure is from the position of release factor GGQ loops visualized to date (as illustrated in Fig. 3b), it could also just be a coincidence. Considering that the authors could do experiments to test their hypothesis (see point #2) that would understandably be beyond the scope of this study, I suggest

moving such speculations to the discussion.

B. If the authors really want to push this point, they should test mutations of the GGN sequence. This could be done by performing in vitro peptidyl-tRNA hydrolysis assays or by looking for cellular phenotypes in which nascent proteins are impaired from being released from 60S subunits with the mutations.

C. The authors reference both eRF1 and Vms1/ANKZF1 as factors that use a GGQ motif to mediate peptidyl-tRNA hydrolysis (lines 219-220). However, Vms1/ANKZF1 does not have a strictly conserved GGQ motif (see doi: 10.1038/s41586-018-0022-5 as cited, as well as 10.1038/s41467-018-04564-3), and it is now known that Vms1/ANKZF1 actually cleaves tRNA instead of hydrolyzing the peptidyl-tRNA bond (doi: 10.1038/s41594-019-0211-4 and 10.1016/j.molcel.2018.08.022). In addition, the GGQ motif of release factors is extremely sensitive to mutations (e.g. doi: 10.1017/s135583829999043x and 10.1093/nar/29.19.3982, which includes a GGQ to GGN mutation). To me, there is just as much existing evidence suggesting that this loop does not act like the GGQ motif of release factors as there is supporting such a model.

D. Although I understand that space in the manuscript is limited, the wording surrounding the comparison with other PTC binders remains somewhat imprecise, as many of these prevent peptidyl-tRNA hydrolysis, while the authors use this section generally to push a model, without functional evidence, suggesting that the loop enhances PTC activity.

Overall, my suggestion is to please either move speculations related to this point to the Discussion (and ideally include more agnostic discussion of existing evidence that do or do not support such an idea if the authors want to push it), or to include experiments to support the authors' hypothesis. In its current state and relative to the other parts of the manuscript, I consider this point somewhat overhyped and potentially misleading.

Minor point: line 471, 483 – “protein translation” is not strictly accurate. mRNAs are translated while proteins are synthesized.

Author Rebuttals to First Revision:

Referees' comments:

Referee #1 (Remarks to the Author):

The authors have diligently addressed my previous comments, enhancing the clarity and robustness of the manuscript. I appreciate their efforts in refining the structure and ensuring the integrity of their work. I do have a few small additional suggestions/ requests I would ask the authors to implement.

1. In the ligase-only structure, there is a minor density blob outside the ligase (see attached image) that seems to originate from the mask, and doesn't contain meaningful density. I suggest that removing this might enhance the quality of the map. Please re-refine without this meaningless blob. I am sure it will improve the overall map quality further.

We thank the reviewer for this suggestion. We have tried re-refining the density with a new mask which excludes this region of the L1 stalk. Unfortunately, the resolution and overall map quality did not improve, likely due to the nature of 3DFlex refinement where the particle alignments are not treated as rigid (**Fig. 1**). Although the new map has less “blob” density, we have decided to use our previous maps and model to avoid having to re-build and refine the structure to match the new map, as the position of the catalytic region has moved slightly during 3DFlex re-refinement.

Figure 1 – Comparison of previous ligase map and re-processed ligase map

a, Previous ligase map and FSC curve after 3DFlex refinement which includes blob of density which corresponds to part of the L1 stalk (red arrow).

b, Re-refined ligase map and FSC curve after 3DFlex training and refinement using a mask that excludes the part of the L1 stalk that contributed to the blob shown in **(a)**. Red arrow points to density of L1 stalk that remains.

2. The clashes depicted in Figure 1f aren't immediately evident from the visual representation provided. The UFBP1 helix appears to fit well into the sec61 channel in the given image. I recommend using surface rendering for both components to clarify the point of contention. Additionally, quantifying the overlap could offer a clearer perspective.

We have generated a new panel (now shown in Figure 1d) indicating the site of the clash. Using surface rendering for both components does help, however, it obstructs the vision and diminishes clarity for the other two panels in Fig. 1d, since these will need to be displayed in the same way for consistency. Instead, we have chosen a new display angle and together with the arrow indicating the clash we feel this now provides a clear perspective.

We also considered quantifying the overlap, however, the maps for the UFBP1 helix are poor and while the helix position is clear, we are not certain on the position of the side chains in this region. Therefore, to avoid providing a misleading figure we prefer to not cite a number.

3. The inclusion of angular distribution plots, which are standard in the cryo-EM field, would be beneficial.

These are now shown in Extended Data Fig. 2.

4. In line with best scientific practices, I suggest the authors make their raw data available on EMPIAR.

We plan to make these available on EMPIAR and are currently transferring the raw data.

Outside of this, I highly recommend this manuscript to be published in Nature!
We thank the reviewer for their support and their suggestions during peer review, which have considerably improved our manuscript.

Referee #2 (Remarks to the Author):

The revised manuscript addresses most of my original concerns. My remaining comment relates, still, to the interpretation of the GGN motif, which I continue to find to be somewhat imprecise and overly speculative in the absence of do-able experiments.

We thank the reviewer for their considered and helpful comments. To avoid speculation and overhyping the role of the GGN motif we have decided to move Figure 3 (focused entirely on PTC loop and GGN motif) to Extended Data Fig. 4. We hope this and the specific comments below satisfy the reviewer's remaining concerns.

Specifically:

A. To my knowledge, the exact mechanism of how the GGQ motif of release factors catalyzes peptidyl-tRNA hydrolysis is not fully understood at a molecular level. It is thought that this motif works with rRNA bases of the peptidyl-transferase center (PTC) to coordinate a water molecule that can act as a nucleophile to hydrolyze the peptidyl-tRNA ester bond. I agree it is tempting to speculate that what the authors modeled may represent some sort of intermediate (e.g. post-hydrolysis) of this process. However, considering how far away the GGN sequence in their structure is from the position of release factor GGQ loops visualized to date (as illustrated in Fig. 3b), it could also just be a coincidence. Considering that the authors could do experiments to test their hypothesis (see point #2) that would understandably be beyond the scope of this study, I suggest moving such speculations to the discussion.

We have now moved the comparison of the PTC loop with the eRF1 GGQ motif to the discussion section.

B. If the authors really want to push this point, they should test mutations of the GGN sequence. This could be done by performing in vitro peptidyl-tRNA hydrolysis assays or by looking for cellular phenotypes in which nascent proteins are impaired from being released from 60S subunits with the mutations.

C. The authors reference both eRF1 and Vms1/ANKZF1 as factors that use a GGQ motif to mediate peptidyl-tRNA hydrolysis (lines 219-220). However, Vms1/ANKZF1 does not have a strictly conserved GGQ motif (see doi: 10.1038/s41586-018-0022-5 as cited, as well as 10.1038/s41467-018-04564-3), and it is now known that Vms1/ANKZF1 actually cleaves tRNA instead of hydrolyzing the peptidyl-tRNA bond (doi: 10.1038/s41594-019-0211-4 and 10.1016/j.molcel.2018.08.022). In addition, the GGQ motif of release factors is extremely sensitive to mutations (e.g. doi: 10.1017/s135583829999043x and 10.1093/nar/29.19.3982, which includes a GGQ to GGN mutation). To me, there is just as much existing evidence suggesting that this loop does not act like the GGQ motif of release factors as there is supporting such a model.

The comparison with Vms1/ANKZF1 has now been removed from the manuscript.

D. Although I understand that space in the manuscript is limited, the wording surrounding the comparison with other PTC binders remains somewhat imprecise, as many of these prevent

peptidyl-tRNA hydrolysis, while the authors use this section generally to push a model, without functional evidence, suggesting that the loop enhances PTC activity. Overall, my suggestion is to please either move speculations related to this point to the Discussion (and ideally include more agnostic discussion of existing evidence that do or do not support such an idea if the authors want to push it), or to include experiments to support the authors' hypothesis. In its current state and relative to the other parts of the manuscript, I consider this point somewhat overhyped and potentially misleading.

We thank the reviewer for the suggestion and as we do not have any experimental evidence to support these claims, we have toned down our claims and also moved the speculation about a possible function of the motif to Discussion.

Minor point: line 471, 483 – “protein translation” is not strictly accurate. mRNAs are translated while proteins are synthesized.

Done